# Error bounds and dynamics of bootstrapping in actor-critic reinforcement learning

**Ahmed J. Zerouali**                                    *ahmed.zerouali@mail.utoronto.ca*
*Department of Physiology*
*University of Toronto*

**Douglas B. Tweed**                                    *tweed.douglas@utoronto.ca*
*Department of Physiology*
*University of Toronto*

**Reviewed on OpenReview:** *https://openreview.net/forum?id=QCjMJfSnYk*

## Abstract

Actor-critic algorithms such as DDPG, TD3, and SAC, which are built on Silver's deterministic policy gradient theorem, are among the most successful reinforcement-learning methods, but their mathematical basis is not entirely clear. In particular, the critic networks in these algorithms learn to estimate action-value functions by a "bootstrapping" technique based on Bellman error, and it is unclear why this approach works so well in practice, given that Bellman error is only very loosely related to *value error*, i.e. to the inaccuracy of the action-value estimate. Here we show that policy training in this class of actor-critic methods depends not on the accuracy of the critic's action-value estimate but on how well the critic estimates the *gradient* of the action-value, which is better assessed using what we call *difference error*. We show that this difference error is closely related to the Bellman error — a finding which helps to explain why Bellman-based bootstrapping leads to good policies. Further, we show that value error and difference error show different dynamics along on-policy trajectories through state-action space: value error is a low-pass anticausal (i.e., backward-in-time) filter of Bellman error, and therefore accumulates along trajectories, whereas difference error is a *high*-pass filter of Bellman error. It follows that techniques which reduce the high-frequency Fourier components of the Bellman error may improve policy training even if they increase the actual size of the Bellman errors. These findings help to explain certain aspects of actor-critic methods that are otherwise theoretically puzzling, such as the use of policy (as distinct from exploratory) noise, and they suggest other measures that may improve these methods.

## 1 Introduction

Actor-critic methods (Witten, 1977), (Barto et al., 1983) are a class of reinforcement-learning algorithms that work well in many applications, especially in continuous control tasks, where a simulated animal or robot learns a motor behavior, such as hopping, walking, or running, based on information from sensors in its body. Recent examples of these methods are DDPG (Lillicrap et al., 2015) and several algorithms that were developed from it, including TD3 (Fujimoto et al., 2018) and SAC (Haarnoja et al., 2018). These latter two methods in particular have been very successful, matching or outperforming all rivals on several benchmark tasks in OpenAI Gym and DeepMind Control Suite. But despite their success, these algorithms are not yet fully understood mathematically. In particular, there are open questions regarding the "bootstrapping" technique that is central to their operation, and which consists of using Bellman error to train a function approximator – the critic – to estimate the action-value function.

The central problem we will address is a paradox concerning this use of the Bellman error. As argued by Fujimoto et al. (2022) among others (Lillicrap et al., 2015), (Fujimoto et al., 2018), Bellman error is

not closely related to the accuracy of the critic – it can be small even when the critic gives a very poor approximation to the true action-value. Nonetheless, critics trained to minimize the squared Bellman error do turn out well, in the sense that they can be used to train good policies. The purpose of this paper is to present mathematical results that partially resolve this paradox, and also help explain other aspects of actor-critic methods that are otherwise puzzling from a theoretical viewpoint.

The remainder of this paper is organized as follows. In section 2, we briefly review actor-critic methods, specify which algorithms we will focus on, and formulate the precise questions that we address in this paper. In section 3, we present our main contributions:

(a) We argue that the pointwise accuracy or value error of the critic, which has been the focus of previous analyses, is not a particularly good index of its ability to train policies. We introduce a more relevant measure called the difference error, which evaluates the directional derivative of the critic's estimates along trajectories, and we relate this new measure to both the Bellman and value errors.

(b) We show that the difference error and the value error are complementary filters of the Bellman error.

(c) We establish a bound on the discrete Fourier coefficients of the Bellman error, which explains how the dynamics of bootstrapping are governed by the regularity of the state dynamics, the policy, the rewards and the critic.

And in section 4, we summarize our results and comment on how they could be built upon in future work.

## 2 Background

### 2.1 Reinforcement learning

The mathematical setting is reinforcement-learning problems where time advances in discrete steps. Let $\mathcal{S} \subseteq \mathbb{R}^d$ and $\mathcal{A} \subseteq \mathbb{R}^p$ denote the *state space* and the *action space* respectively. At each time $t$, the agent receives information about the current state $s_t \in \mathcal{S}$ of its environment (and we will focus on the case where that information is complete and accurate, or in other words where the state is fully observable). The agent then applies a function $\mu : \mathcal{S} \to \mathcal{A}$, called the *actor* or the *policy*, to choose an *action*, $a_t = \mu(s_t) \in \mathcal{A}$. Having made this choice, the agent gets a scalar reward $r_t = r(s_t, a_t) \in \mathbb{R}$ typically depending on both the state and action; we assume that the function $r : \mathcal{S} \times \mathcal{A} \to \mathbb{R}$ is bounded, as in most real-world applications. Time ticks forward to $t + 1$, and the environment passes to its next state, $s_{t+1} = f(s_t, a_t)$, where $f$ is the *state transition* or *state dynamics* function. The reward and dynamics functions are deterministic, and the policy is also deterministic apart from "exploratory" and "policy" noise terms, described below, that are added to its outputs during training but not during testing. In recent applications, the policy is most often a deep neural network, and the aim of the reinforcement-learning algorithm is to adjust the weights and biases of that network to yield an optimal policy, or in other words one that maximizes the discounted cumulative reward or *value*,

$$V^\mu(s_t) \triangleq \sum_{\tau=t}^{\infty} \gamma^{\tau-t} r\left(s_\tau, \mu(s_\tau)\right), \tag{2.1}$$

averaged across all possible starting points $s_t$ of trajectories in state space. In this formula, $\gamma$ is a "discount factor" in the range $(0, 1)$ which expresses the idea that rewards in the distant future matter less to the agent than more imminent rewards do. More generally, the final time point in the summation need not be $\infty$, but we will assume that it is, to simplify the math. So in short, the aim is to adjust $\mu$ to maximize $V^\mu$.

Actor-critic methods approach this problem by creating a *critic*, which is a function approximator that is distinct from the policy network, and is trained to learn the *action-value* function $Q^\mu : \mathcal{S} \times \mathcal{A} \to \mathbb{R}$, which takes as input a state-action pair $(s, a)$ and yields as output the quality of its outcome – the total discounted cumulative reward that will result from taking action $a$ in state $s$ and then choosing all subsequent actions

in accordance with policy $\mu$:

$$Q^\mu(s_t, a_t) \triangleq r(s_t, a_t) + \sum_{\tau=t+1}^{\infty} \gamma^{\tau-t} r\left(s_\tau, \mu(s_\tau)\right). \tag{2.2}$$

Clearly $Q^\mu$ is closely related to $V^\mu$, as $Q^\mu(s, \mu(s)) = V^\mu(s)$, again assuming (as we do throughout the paper) that the dynamics, reward, and policy are deterministic.

From (2.2) it follows that the action-value function obeys the *Bellman equation* (Bellman, 1957; Sutton & Barto, 2018),

$$Q^\mu(s_t, a_t) = r(s_t, a_t) + \gamma Q^\mu\left(s_{t+1}, \mu(s_{t+1})\right), \tag{2.3}$$

which we can write more simply as

$$Q_t^\mu = r_t + \gamma Q_{t+1}^\mu, \tag{2.4}$$

where we use the shorthand notation $Q_t^\mu = Q^\mu(s_t, a_t)$, $r_t = r(s_t, a_t)$, and $Q_{t+1}^\mu = Q^\mu\left(s_{t+1}, \mu(s_{t+1})\right)$. Note that the action $a_t$ at time $t$ is arbitrary while the action at time $t+1$ must be on policy: $a_{t+1} = \mu(s_{t+1})$.

Usually, the critic $Q : \mathcal{S} \times \mathcal{A} \to \mathbb{R}$ is trained by adjusting its parameters to shrink the *Bellman error* $e^B : \mathcal{S} \times \mathcal{A} \to \mathbb{R}$, given at $(s_t, a_t)$ by

$$e^B(s_t, a_t) \triangleq Q(s_t, a_t) - r(s_t, a_t) - \gamma Q\left(s_{t+1}, \mu(s_{t+1})\right), \tag{2.5}$$

or more briefly by

$$e_t^B \triangleq Q_t - r_t - \gamma Q_{t+1}, \tag{2.6}$$

with the same shorthand notation as in the previous paragraph.

What is noteworthy here is that $Q$ is adjusted based on its own values at two time points, $Q_t$ and $Q_{t+1}$ – a process called *bootstrapping* – rather than on any direct feedback about $Q^\mu$, the function it is trying to approximate. In effect, $Q$ is adjusted to obey more and more closely the Bellman equation (2.4), in the hope that, if both $Q$ and $Q^\mu$ obey that equation, then $Q$ may resemble $Q^\mu$ in other ways as well. But in which respects, exactly, do we need to make $Q$ resemble $Q^\mu$, and what is the mathematical justification for hoping that this resemblance can be achieved by minimizing the Bellman error?

In addressing these questions, we will restrict our discussion to the large subclass of actor-critic methods that are *off policy*, that work with *continuous* state and action spaces, and that rely on the Silver et al. (2014) *deterministic policy gradient* (DPG) theorem – a result that mathematically justifies the procedure of improving the policy network by adjusting its parameters $\theta^\mu$ up the gradient calculated by the chain rule:

$$\left.\frac{\partial Q^\mu(s, a)}{\partial \theta^\mu}\right|_{(s,a)=(s,\mu(s))} = \left(\left.\frac{\partial Q^\mu(s, a)}{\partial a}\right|_{(s,a)=(s,\mu(s))}\right) \cdot \frac{\partial \mu(s)}{\partial \theta^\mu}. \tag{2.7}$$

(More precisely, the DPG theorem shows that, in our continuous-control setting, $\partial Q/\partial \theta^\mu$ approximates the gradient of a reasonable objective function for the policy – see (Degris et al., 2012), (Silver et al., 2014), (Imani et al., 2018)).

This subclass of algorithms includes many important actor-critic algorithms such as DDPG (Lillicrap et al., 2015), TD3 (Fujimoto et al., 2018), and SAC (Haarnoja et al., 2018), but excludes other highly successful actor-critic methods such as PPO (Schulman et al., 2017).

The justification for bootstrapping is less of a problem for PPO than for off-policy DPG-based algorithms, which is why we focus on the latter class here. To clarify this point, we note that PPO estimates $Q^\mu$ directly by summing discounted rewards along its most recent trajectory, and the critic network supplies a "baseline" or "control variate" to reduce the variance of the direct estimate. In this setup, it is well understood that even a very poor estimate of $Q^\mu$ can serve as a useful control variate, provided that it correlates with the true $Q^\mu$. In DPG-type algorithms however, the critic network $Q$ is the policy's sole source of information about $Q^\mu$, and so inaccurate critics are a more severe problem, and the need to justify bootstrapping is more pressing.

Focusing our discussion on off-policy DPG-based continuous-control algorithms also motivates our restriction to deterministic dynamics. In practice, the performance of these algorithms is commonly evaluated on deterministic tasks and environments such as HalfCheetah, Walker2d, Ant, Humanoid and so on in Mujoco and the OpenAI Gym/Gymnasium package and the DeepMind Control Suite. On a more conceptual level, it is better to analyse the simpler, deterministic case first, and extend the results to stochastic systems in a separate paper, as the mathematical treatment would be more involved and potentially less instructive.

### 2.2 Question

Why do DPG-based actor-critic methods perform so well in practice, given that they rely on an estimator of $Q^\mu$ that is trained using an error signal, $e^B$, which is computed by bootstrapping, with no reference to $Q^\mu$?

The usual justification is a theorem of (Bertsekas & Tsitsiklis, 1996) which guarantees that if $e^B = 0$ for all state-action pairs $(s, a)$, then the estimator $Q$ will be perfectly accurate, meaning that what we will call the *value error* is zero,

$$e^Q(s, a) \triangleq Q(s, a) - Q^\mu(s, a) = 0, \tag{2.8}$$

for all $(s, a) \in \mathcal{S} \times \mathcal{A}$.

Yet this standard rationale is not entirely reassuring, because it applies only in cases where $e^B$ is exactly zero throughout state-action space, whereas in practice $e^B$ is never zeroed, and the best we can hope for is that it will be small enough. Worryingly in this context, it has been shown by (Fujimoto et al., 2022) that if $e^B$ is not zeroed but merely bounded, then $e^Q$ may be very large even when $e^B$ is small. Here we analyze further the relation between $e^B$ and action-value estimators in the setting of DPG-based actor-critic methods.

In the proofs that follow, we analyse the information about $\partial Q^\mu / \partial a$ provided by a single sample or minibatch of Bellman errors. We make no assumptions about how the critic or policy have been trained before that batch was drawn, or whether they have been trained at all. Our results hold even if the critic and policy are freshly initialized, with random parameters. Similarly, we will analyse the evolution of various error measures along infinite-duration trajectories, but we do not assume that the agent has traversed those trajectories, or any trajectories at all. The point is that, even if the agent has never had any encounter of any kind with its environment, still its critic and policy and the state dynamics and reward function together imply a continuum of infinite-duration trajectories filling the on-policy submanifold of state-action space, and they also imply the errors that the agent would encounter if it did traverse any portion of any of those paths, or if it computed Bellman and other errors based on tuples drawn from a replay buffer.

## 3 Main results

### 3.1 Critics compute partial derivatives

As mentioned in the previous section, DPG-based actor-critic algorithms rely on the gradient $\partial Q / \partial \theta^\mu = \partial Q / \partial a \cdot \partial \mu / \partial \theta^\mu$ to adjust the policy parameters $\theta^\mu$ of $\mu : \mathcal{S} \to \mathcal{A}$. Therefore, the critic learns $Q^\mu$ *only* in order to compute an estimate of the partial derivative $\partial Q^\mu / \partial a$. This is a key fact, because it means that during learning, *the accuracy of the approximator $Q$ itself matters less than that of the gradient estimate $\partial Q / \partial a$.* And therefore the value error $e^Q$ is a less relevant measure of the critic's quality than what we will call the *difference error*,

$$e_t^{\Delta Q} \triangleq (Q_{t+1} - Q_t) - (Q_{t+1}^\mu - Q_t^\mu) = e_{t+1}^Q - e_t^Q = \Delta e_t^Q. \tag{3.1}$$

Simply put, $e_t^{\Delta Q}$ is a better gauge than $e_t^Q$ of the critic's gradient estimate because the gradient depends on how $Q$ and $Q^\mu$ *change* from point to point across state-action space, not on their values at any one point. To express the same idea mathematically, we define the *state-action step* $\Delta x_t = (s_{t+1} - s_t, \mu(s_{t+1}) - a_t)$, and consider the first-order Taylor expansion

$$Q_t = Q_{t+1} - \Delta x_t^\intercal \cdot \nabla Q(s_{t+1}, \mu(s_{t+1})) + \mathrm{O}\left(\|\Delta x_t\|^2\right)$$

for the approximator $Q$, as well as a similar expression for the true action-value function $Q^\mu$. Assuming that $\|\Delta x_t\|^2$ is small enough, we can write that

$$
\begin{aligned}
e_t^{\Delta Q} &\approx \Delta x_t^\intercal \cdot \left( \nabla Q\left(s_{t+1}, \mu(s_{t+1})\right) - \nabla Q^\mu\left(s_{t+1}, \mu(s_{t+1})\right) \right) \\
&\triangleq \Delta x_t^\intercal \cdot e_{t+1}^{\nabla Q},
\end{aligned}
\tag{3.2}
$$

where $e_{t+1}^{\nabla Q}$ is the critic's *gradient error*. This equation shows that $e_t^{\Delta Q}$ contains information about the gradient error which $e_t^Q$ does not. Namely, $e_t^{\Delta Q}$ is *approximately the inner product of $e_t^{\nabla Q}$ with the vector $\Delta x_t$*, or in other words *it approximates the error in the directional derivative of the critic's estimates.*

To summarize, (Fujimoto et al., 2022) and others have raised doubts about Bellman-based learning, on the grounds that $e_t^B$ is a poor proxy for $e_t^Q$. Our plan is to defend Bellman methods by showing that $e_t^B$ is a very good proxy for the more relevant error measure $e_t^{\Delta Q}$. One could of course study the relation between $e_t^B$ and the gradient error $e_t^{\nabla Q}$ itself, or even the *action-gradient error* given by

$$
e^{\partial_a Q}(s, a) \triangleq \frac{\partial Q}{\partial a}(s, a) - \frac{\partial Q^\mu}{\partial a}(s, a),
$$

but those relations are complicated, in part because they depend on the detailed contents of the replay buffer (see Appendix A). We have found it more informative to establish the relatively simple relation between the Bellman and difference errors and then, in Appendix A, build on equation (3.2) to describe the relation between $e_t^{\Delta Q}$ and the gradient errors. There we will further highlight the suitability of $e_t^{\Delta Q}$ as an index of gradient error by showing that it is approximately linearly related to the best estimate of $e_{t+1}^{\partial_a Q}$ that can be obtained based on the information in the replay buffer.

**Proposition 1.** *The difference error is related to the Bellman error (2.6) and the value error (2.8) by the equation*

$$
e_t^{\Delta Q} = -\frac{e_t^B}{\gamma} + \frac{(1-\gamma)}{\gamma} e_t^Q.
\tag{3.3}
$$

*Proof.* We subtract the right-hand side of (3.3) from the left, multiply by $\gamma$, expand based on the definitions of $e_t^{\Delta Q}$, $e_t^B$, and $e_t^Q$, and then simplify, to get

$$
\begin{aligned}
\gamma e_t^{\Delta Q} + e_t^B - (1-\gamma)e_t^Q &= \gamma(Q_{t+1} - Q_t - Q_{t+1}^\mu + Q_t^\mu) + (Q_t - r_t - \gamma Q_{t+1}) - (1-\gamma)(Q_t - Q_t^\mu) \\
&= Q_t^\mu - r_t - \gamma Q_{t+1}^\mu = 0,
\end{aligned}
$$

where the final equality follows from Bellman's equation. $\qquad\square$

With (3.3) in hand, we can compare how $e_t^Q$ and $e_t^{\Delta Q}$ are bounded in relation to $e_t^B$.

Regarding $e_t^Q$, Fujimoto et al. (2022) have shown that if the Bellman error is bounded so that $|e_t^B| \leq C$ for some constant $C > 0$, then

$$
|e_t^Q| \leq \frac{C}{(1-\gamma)}.
\tag{3.4}
$$

That is, $|e_t^Q|$ may be as large as $C/(1-\gamma)$, for example 100 times larger than $C$ if $\gamma = 0.99$, which is a common value used in many applications (Lillicrap et al., 2015; Fujimoto et al., 2018; Haarnoja et al., 2018).

But the corresponding bound for $e_t^{\Delta Q}$ is much better:

**Proposition 2.** *If $|e_t^B| \leq C$, then*

$$
|e_t^{\Delta Q}| \leq \frac{2C}{\gamma}.
\tag{3.5}
$$

*Proof.* From equations (3.3) and (3.4),

$$
|e_t^{\Delta Q}| \leq \frac{|-e_t^B|}{\gamma} + \frac{(1-\gamma)}{\gamma}|e_t^Q| \leq \frac{C}{\gamma} + \frac{(1-\gamma)}{\gamma}\frac{C}{(1-\gamma)} = \frac{2C}{\gamma}.
$$

$\qquad\square$

This bound is of particular importance. Again, for the concrete value $\gamma = 0.99$ often used in practice, we see that $|e^{\Delta Q}|$ is at worst just over twice as large as the upper-bound on $|e^B|$, whereas $|e^Q|$ can be as much as two orders of magnitude larger than $|e^B|$. In short, while $e^B$ is a poor proxy for $e^Q$, it is a good proxy for $e^{\Delta Q}$. And $e^{\Delta Q}$ is the more relevant measure: in light of the discussion at the beginning of this subsection, $e^{\Delta Q}$ better reflects the accuracy of the critic's gradient estimate $\partial Q/\partial a$, which is what matters for policy training. Therefore the bound (3.5) *helps explain why DPG-based actor-critic learning works well in practice*, as training $Q$ to shrink $e^B$ indirectly improves the approximation of $\partial Q^\mu/\partial a$ that is used for policy updates.

Of course, this finding does not resolve all the mathematical questions regarding actor-critic methods, and $e^{\Delta Q}$ is not a *perfect* substitute for $e^{\partial_a Q}$ or $e^{\nabla Q}$, as we explain in Appendix A. However, more can be said once we clarify the underlying reason why $e^Q$ and $e^{\Delta Q}$ have such different bounds, which we do in the next subsection.

## 3.2  $e_t^Q$ and $e_t^{\Delta Q}$ are complementary filters

Here we show that $e_t^Q$ and $e_t^{\Delta Q}$ are inversely related, in the sense that $e_t^Q$ is a *low-pass filter of $e_t^B$, whereas $e_t^{\Delta Q}$ is a high-pass filter of $e_t^B$*. Therefore it is not just the size of $e_t^B$ that influences $e_t^Q$ and $e_t^{\Delta Q}$. Rather, a crucial factor is the *temporal frequency* of $e_t^B$ along the trajectories of the system, by which we mean the frequency of variation of $e_t^B$ with respect to $t$. It follows that techniques that reduce this temporal frequency may improve the performance of actor-critic methods even if they increase $e_t^B$ itself.

In our discrete-time setting, a first-order linear time-invariant low-pass filter, or more simply a *low-pass filter* from now on, can be described by the equation

$$y_{t+1} = \alpha x_{t+1} + (1 - \beta)y_t. \tag{3.6}$$

where $x_t$ is the filter's input, $y_t$ is its output, and $\alpha$ and $\beta$ are positive constants (Oppenheim et al., 1998, Sec.3.9-11). The *gain* of this filter is $\alpha/\beta$, which means that, given a constant input $x_t = x$, the filter's output $y_t$ will eventually converge to a steady, equilibrium value of $(\alpha/\beta)x$. (In some papers, low-pass filters are defined to have $\alpha = \beta$, and therefore a gain of 1, in which case any non-unity scaling is applied afterwards by multiplying the filter output by the desired gain factor, but for us it will be convenient to treat the gain as an intrinsic property of the filter.)

Returning now to the Bellman equation (2.4), we can write it this way:

$$Q_t^\mu = r_t + [1 - (1 - \gamma)] \, Q_{t+1}^\mu, \tag{3.7}$$

which has the same form as (3.6), except that the time indices $t$ and $t + 1$ have been swapped. In other words, the Bellman equation defines a filter running backwards in time, or more briefly, an *anticausal* filter. Therefore we have:

**Proposition 3.** *The function $Q_t^\mu$ is an anticausal low-pass filter of $r_t$, with a gain of $1/(1 - \gamma)$.*

Similarly, (2.6) can be written

$$Q_t = e_t^B + r_t + [1 - (1 - \gamma)] \, Q_{t+1}. \tag{3.8}$$

Subtracting (3.7) from (3.8) gives us:

**Proposition 4.** *The value error $e_t^Q$ is an anticausal low-pass filter of the Bellman error $e_t^B$ with constants $\alpha = 1$ and $\beta = (1 - \gamma)$:*

$$e_t^Q = e_t^B + [1 - (1 - \gamma)] \, e_{t+1}^Q, \tag{3.9}$$

*and so we have*

$$e_t^Q = \sum_{\tau=t}^{\infty} \gamma^{(\tau-t)} e_\tau^B. \tag{3.10}$$

*Proof.* We establish equation (3.10). From equation (3.9) we have

$$e_\tau^B = e_\tau^Q - \gamma e_{\tau+1}^Q, \quad \text{for all } \tau \geq t,$$

so by multiplying both sides of this equality by $\gamma^{(\tau-t)}$ and then summing over $\tau = t + k$ with $k = 0, 1, \cdots$, we get

$$\sum_{\tau=t}^{\infty} \gamma^{(\tau-t)} e_{\tau}^{B} = \sum_{\tau=t}^{\infty} \gamma^{(\tau-t)} e_{\tau}^{Q} - \sum_{\tau=t}^{\infty} \gamma^{(\tau+1-t)} e_{\tau+1}^{Q}$$
$$= \sum_{\tau=t}^{\infty} \gamma^{(\tau-t)} e_{\tau}^{Q} - \sum_{\tau=t+1}^{\infty} \gamma^{(\tau-t)} e_{\tau}^{Q}$$
$$= e_{t}^{Q}.$$

$\square$

The gain of this filter is $1/(1-\gamma)$ which in practice is usually large. For instance, if $\gamma = 0.99$ then the gain is 100. *This high-gain, low-pass filter behavior is the reason $e_t^Q$ can grow so much larger than $e_t^B$.*

In this same discrete-time setting, a *high*-pass filter (Oppenheim et al., 1998, Sec.3.10) is described by the equation

$$y_{t+1} = \alpha(x_{t+1} - x_t) + (1 - \beta)y_t. \tag{3.11}$$

From this fact, together with the definition of $e_t^{\Delta Q}$ in (3.1), and equation (3.10), it follows that:

**Proposition 5.** *The difference error $e_t^{\Delta Q}$ is an anticausal high-pass filter of the Bellman error $e_t^B$ with $\alpha = 1$ and $\beta = 1 - \gamma$:*

$$e_t^{\Delta Q} = \left(e_{t+1}^B - e_t^B\right) + \left[1 - (1-\gamma)\right] e_{t+1}^{\Delta Q}, \tag{3.12}$$

*and so*

$$e_t^{\Delta Q} = \sum_{\tau=t}^{\infty} \gamma^{(\tau-t)} \left(e_{\tau+1}^B - e_{\tau}^B\right). \tag{3.13}$$

*Proof.* We have

$$e_t^{\Delta Q} = \left(Q_{t+1} - Q_{t+1}^{\mu}\right) - \left(Q_t - Q_t^{\mu}\right) = \left(Q_{t+1} - r_{t+1} - \gamma Q_{t+2}^{\mu}\right) - \left(Q_t - r_t - \gamma Q_{t+1}^{\mu}\right)$$
$$= (Q_{t+1} - r_{t+1} - \gamma Q_{t+2}) - (Q_t - r_t - \gamma Q_{t+1}) + \gamma \left(Q_{t+2} - Q_{t+1} - Q_{t+2}^{\mu} + Q_{t+1}^{\mu}\right)$$
$$= \left(e_{t+1}^B - e_t^B\right) + \gamma e_{t+1}^{\Delta Q}$$

to establish (3.12). Equation (3.13) then follows by the same reasoning as that of (3.10). $\square$

From this result, we know that $e_t^{\Delta Q}$ shows the characteristic behavior of high-pass filters (Oppenheim et al., 1998): it ignores low-frequency events, and it responds to high-frequency events but then "forgets", its value fading to zero with the (backwards) passage of time.

So the main point of this section is that, owing to their different filtering properties, *$e_t^Q$ accumulates along trajectories whereas $e_t^{\Delta Q}$ does not*, and this is the underlying reason that the bounds on $e_t^Q$ discovered by Fujimoto et al. (2022) are large whereas the bounds on $e_t^{\Delta Q}$ are small. Pushing this analysis further, we have:

**Corollary 6.** *Let $e^B : \mathcal{S} \times \mathcal{A} \to \mathbb{R}$ be upper-bounded by $C > 0$, and suppose that there exists a point $(s_0, a_0) \in \mathcal{S} \times \mathcal{A}$ at which*

$$e^Q(s_0, a_0) = \frac{C}{1 - \gamma}.$$

*Then along the trajectory starting at $(s_0, a_0)$ and following the policy $\mu$, we have that $e_t^{\Delta Q} = 0$ for all $t \in \mathbb{N}$.*

*Proof.* As $|e^B| \leq C$ and the anticausal low-pass filter $e_t^Q$ has a gain of $1/(1-\gamma)$, the time series $e_t^Q$ can attain a value of $C/(1-\gamma)$ if and only if it receives a strictly constant input $e_t^B = C$ for all $t \in \mathbb{N}$. The vanishing of $e_t^{\Delta Q}$ along the trajectory $\{(s_0, a_0), (s_t, \mu(s_t))\}_{t=1}^{\infty}$ then follows from equation (3.13). $\square$

In words, if the Fujimoto et al. bound (3.4) is tight at any point in state-action space, then the difference error vanishes everywhere along the trajectory through that point. In that sense, there is a partial trade-off between $e_t^Q$ and $e_t^{\Delta Q}$: for any given bound on $e_t^B$, the absolute value $|e_t^Q|$ reaches its maximum only when $|e_t^{\Delta Q}|$ is minimal (i.e. 0). Another practical consequence, more clearly expressed by equation (3.13), is that for any given magnitude of the Bellman error, the lower we can make the temporal frequency of variation of $e_t^B$, the smaller $|e_t^{\Delta Q}|$ will be. We elaborate on this point in the next subsection.

### 3.3 Controlling temporal frequency

What factors influence the temporal frequency of $e_t^B$? The standard way of analyzing frequency components of time series is to use the discrete Fourier transform, or DFT: given any finite-length real time series $\{x_n\}_{n=0}^{N-1} = \{x_0, x_1, \cdots, x_{N-1}\}$, where $N$ is a positive integer, the DFT of $\{x_n\}_{n=0}^{N-1}$ is the $N$-element sequence $\{\hat{x}_k\}_{k=0}^{N-1}$ given by (Stankovic, 2015)

$$\hat{x}_k = \sum_{n=0}^{N-1} x_n \exp(-\mathrm{i}2\pi kn/N), \quad k = 0 \cdots, (N-1).$$

Here, the modulus $|\hat{x}_k|$ corresponds to $N$ times the amplitude of the component of $\{x_n\}_{n=0}^{N-1}$ of frequency $2\pi k/N$. Please see Appendix B for more explanations of the DFT and a proof of the following bound on the frequency components of the Bellman error:

**Proposition 7.** *Along any finite-length segment $\{(s_0, a_0), (s_n, \mu(s_n))\}_{n=1}^{N-1}$ of a trajectory, the DFT terms $\{\hat{e}_k^B\}_{k=1}^{N-1}$ of the Bellman error satisfy the inequality*

$$\left|\hat{e}_k^B\right| \leq \frac{(N-2)}{\sin\left(\frac{\pi k}{N}\right)} \left\{ \|f_\mu^\Delta\|_{\max} \sqrt{1 + \mathrm{Lip}(\mu)^2} \left[(1+\gamma)\mathrm{Lip}(Q) + \mathrm{Lip}(r)\right] \right\}$$
$$+ \left[(1+\gamma)|Q_1 - Q_0| + |r_1 - r_0|\right] + \gamma\left(|Q_0| + |Q_N|\right),$$

*where $\mathrm{Lip}(r)$, $\mathrm{Lip}(Q)$, and $\mathrm{Lip}(\mu)$ are the Lipschitz constants of $r : \mathcal{S} \times \mathcal{A} \to \mathbb{R}$, $Q : \mathcal{S} \times \mathcal{A} \to \mathbb{R}$, and $\mu : \mathcal{S} \to \mathcal{A}$ respectively, $f_\mu^\Delta(s) \triangleq f(s, \mu(s)) - s$, and where $\|f_\mu^\Delta\|_{\max} = \max_{\mathcal{S}} \|f_\mu^\Delta\|$.*

In simple terms, the bounds on $|\hat{e}_k^B|$ depend on the frequency $2\pi k/N$ through the factor $(N-2)/\sin(\pi k/N)$, as well as on the "smoothness" of the functions $r$ and $Q$ across state-action space, and on the system's motion through that space, which is determined by the policy $\mu$ and the state dynamics $f$. To put this result in context, we argued in section 3.1 that minimizing the difference error $e_t^{\Delta Q}$ is more important than minimizing the $Q$-function error $e_t^Q$, and in section 3.2, we showed that $e_t^{\Delta Q}$ is a high-pass filter of $e_t^B$, meaning that the scale of $|e_t^{\Delta Q}|$ is governed by the absolute values $|\hat{e}_k^B|$ for $k$ near $N/2$ (the high-frequency components). The proposition above now explains how the choices of $Q$, $r$, and $\mu$ will impact the upper-bounds on the $|\hat{e}_k^B|$, in the sense that the less abruptly these functions vary at a local scale, the smaller $|e_t^{\Delta Q}|$ must be.

At this point, it may be helpful to visualize how some of these factors – the regularity of $\mu$, $r$ and $Q$ over $\mathcal{S}$ and $\mathcal{A}$ – are interrelated and how they influence the temporal frequency of $Q$ along an on-policy trajectory $\{(s_t, \mu(s_t))\}_{t \geq 0}$. We focus on $t \mapsto Q(s_t, \mu(s_t))$, because it is one of the main variables that determines the temporal behavior of $e_t^B$. (We could of course have shown $e^B$ itself rather than $Q$, but plots of $e^B : \mathcal{S} \times \mathcal{A} \to \mathbb{R}$ and $t \mapsto e^B(s_t, \mu(s_t))$ are less easy to interpret.)

We will consider a very simple control system, a one-kilogram mass sliding on a horizontal frictionless rail in zero gravity. The state space $\mathcal{S} \subset \mathbb{R}^2$ of this system consists of the position and velocity $s = (q, v)$ of the mass, while the action is a driving force applied to it, so that $\mathcal{A} = \mathbb{R}$. The dynamics function is obtained by discretizing Newton's second law by Euler's method, with time step 1 ms. Figure 3.1 showcases the following:

- Panel (A) represents a reference case. Here, for simplicity, the policy $\mu$ is linear and the reward function is quadratic. Specifically, the policy's action is an applied force, expressed in newtons and given by $\mu(s) = -\frac{1}{2}(v + 5(q - 0.3))$, where $q$ is in metres and $v$ in m/s. The reward is $r(s, a) = -(3 \times 10^{-4})(q^2 + 10v^2)$, which is bounded over the trajectories we consider in the figure.

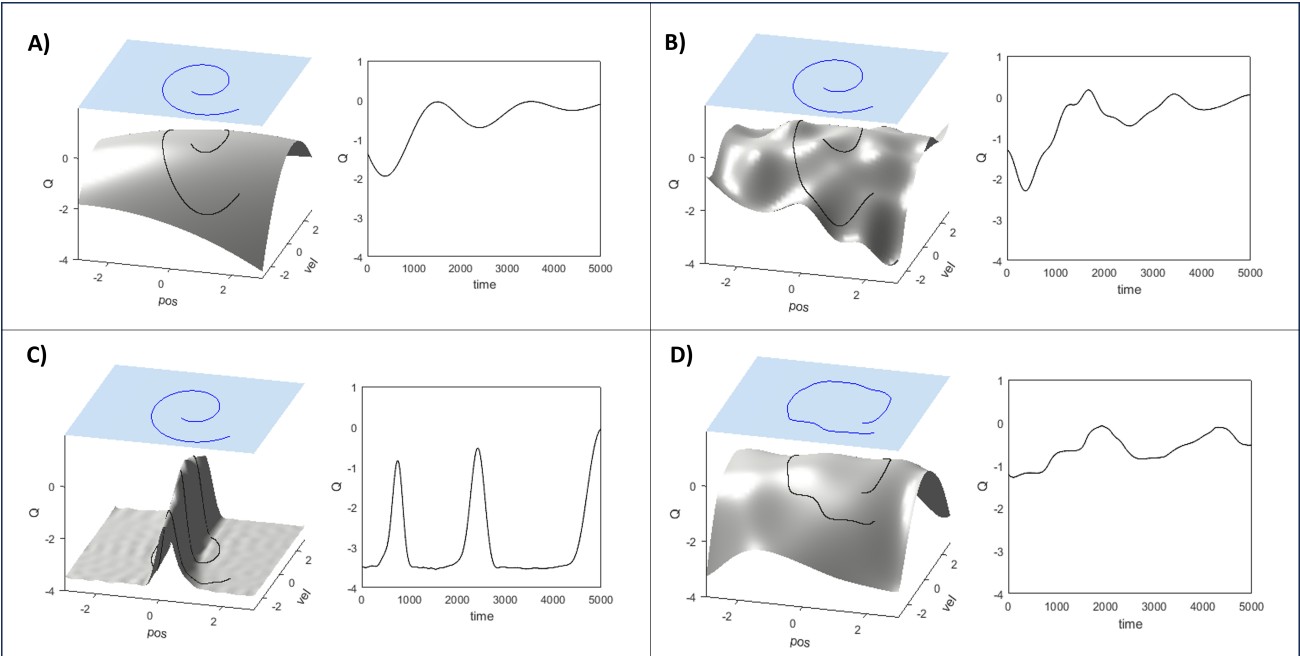

Figure 3.1: Spatial factors affecting temporal frequency along state-space trajectories. *Left of each panel:* on top is state space $\mathcal{S}$ (the blue plane) with one on-policy trajectory; below is the graph of the approximator $Q$ (the grey surface) with the projection of the same trajectory (black curve). *Right of each panel:* a plot of $t \mapsto Q\left(s_t, \mu(s_t)\right)$. *(A)* Low $\mathrm{Lip}(\mu)$ and $\mathrm{Lip}(r)$ with fine approximator $Q$ (reference case); *(B)* Low $\mathrm{Lip}(\mu)$ and $\mathrm{Lip}(r)$ with coarse approximator $Q$; *(C)* Low $\mathrm{Lip}(\mu)$ and high $\mathrm{Lip}(r)$ with fine approximator $Q$; *(D)* Low $\mathrm{Lip}(r)$ and high $\mathrm{Lip}(\mu)$ with fine approximator $Q$.

> $\mathrm{Lip}(\mu)$ and $\mathrm{Lip}(r)$ are both low. The critic $Q$, consisting of 500 Gaussian filters trained by the recursive least squares method (RLS), is an almost perfect fit to the true $Q^\mu$(which we approximated very closely by summing discounted rewards along trajectories). The near-perfect fit and the low $\mathrm{Lip}(r)$ induce a low $\mathrm{Lip}(Q)$, resulting in a $t \mapsto Q\left(s_t, \mu(s_t)\right)$ with few fluctuations.

- Panel (B) depicts the situation where $\mu$ and $r$ are the same as in (A), but the approximator $Q$ has only 50 filters. Its coarser approximation has a higher $\mathrm{Lip}(Q)$ than the reference case, and leads to more fluctuations in the right-hand-side plot.

- Panel (C) is a case where $\mu$ and $Q$ are as in (A), but $r(s, a) = -0.035\tanh(10q^2)$ instead of the reference quadratic. In this example, the larger $\mathrm{Lip}(r)$ induces a larger $\mathrm{Lip}(Q^\mu)$ and therefore a larger $\mathrm{Lip}(Q)$, which can be seen from the steep bumps in the graph of $Q$, again resulting in fluctuations in the right-hand plot.

- Finally, Panel (D) shows a case where $r$ and $Q$ are as in (A), but the linear policy is replaced by a coarse approximation $\mu$. In this case, not only does the trajectory fluctuate more in $\mathcal{S}$ because of the higher $\mathrm{Lip}(\mu)$, but we also obtain a higher $\mathrm{Lip}(Q)$ because the rougher policy affects $Q^\mu$.

To summarize, Fig. 3.1 illustrates how the smoothness of $\mu$, $r$ and $Q$ over their domain spaces influences the temporal frequency of $t \mapsto Q\left(s_t, \mu(s_t)\right)$, whose behavior carries over to the Bellman error, which is computed using $Q$ and $r$.

The main lesson of Proposition 7 and Figure 3.1 is that we can shrink the high-frequency components of $e_t^B$ by reducing $\|f_\mu^\Delta\|_{\max}$, or by choosing a simple, low-Lipschitz reward function $r$, or by smoothing out $\mu$ or $Q$, for instance with weight decay. As regards $\|f_\mu^\Delta\|_{\max}$, it is usually not possible to alter the function $f^\Delta$, which is determined by the state dynamics. But in some tasks it may be possible to train an agent initially in a

simplified, lower-speed version of the environment it will ultimately operate in. This is a sensorimotor form of curriculum learning (Bengio et al., 2009; Wang et al., 2021), and reflects the commonplace observation that humans do often begin learning a skill in a simplified or lower-speed setting, as for instance with training wheels or on a kiddie slope.

### 3.4 Relation to policy noise

Our focus has been on the mathematical justification for Bellman-based learning, but our temporal filtering results also shed light on another aspect of off-policy DPG-type actor-critic methods, namely the use of "policy noise".

In many algorithms of this type, the agent stores a large number of its past interactions with the environment in a "replay buffer", where the $k$th entry in the buffer is a tuplet $(s_k, a_k, r_k, s'_k)$, where $s_k$ was the state at the beginning of an interaction, $a_k$ was the action taken, $r_k$ was the resulting reward, and $s'_k$ was the subsequent state. The critic trains itself by drawing batches of these tuples from the buffer and for each one computing not its associated Bellman error $e^B$ but a perturbed version of that error:

$$\tilde{e}_k^B = Q(s_k, a_k) - r_k - \gamma Q^{\text{tgt}}\left(s'_k, \mu(s'_k) + \nu\right), \tag{3.14}$$

where $\nu$ is a zero-mean *policy noise* term (distinct from the "exploratory noise" that the agent adds to its actions when it interacts with the environment), and where $Q^{\text{tgt}}$ designates a target network, which is close to but not identical with $Q$.

While the motivation behind *exploratory* noise is obvious, and the use of a target net has been shown to bring certain benefits (e.g. in (Fan et al., 2020)), the use of the *policy* noise term $\nu$ in (3.14) is less clear conceptually, as its addition violates the rationale behind the learning rule. Indeed, the rationale for learning from $e^B$ is the Bellman equation (2.3), but that equation holds only when the action at the subsequent state is on policy (i.e. $a_{t+1} = \mu(s_{t+1})$ or, in the buffer, $\mu(s'_k)$); it fails if noise is added to $\mu(s'_j)$ as in (3.14). And yet adding $\nu$ does improve learning. Fujimoto et al. (2018), who introduced the idea of policy noise, proposed that it might help performance by smoothing the learned $Q$ estimate. In light of our results we can add that the spatial averaging induced by $\nu$ of the target value around $Q^{\text{tgt}}(s', \mu(s'))$ may also be useful because it blurs out high-frequency components of $Q$. In other words, our findings here clarify the sense in which using equation (3.14) regularizes the critic's learning, and perhaps provide a fuller explanation as to why policy noise improves actor-critic performance.

## 4 Summary and future work

In this paper we have addressed the question, why do DPG-based actor-critic methods, which train a value-estimator $Q$ based on the Bellman error $e_t^B$, perform well even though $e_t^B$ is not closely related to the value error $e_t^Q$, and $e_t^Q$ can be very large even when $e_t^B$ is small? The answer, we have shown, is that the accuracy of the policy's teaching signal $\partial Q / \partial a$ depends less on the value error $e_t^Q$ than on the difference error $e_t^{\Delta Q}$, and $e_t^{\Delta Q}$ *is* closely related to $e_t^B$. We have also shown that this difference error is a high-pass-filtered version of $e_t^B$, suggesting that actor-critic performance may be improvable by taking steps to limit the high-frequency spatial components of the functions $r : \mathcal{S} \times \mathcal{A} \to \mathbb{R}$ and $Q : \mathcal{S} \times \mathcal{A} \to \mathbb{R}$, for instance by choosing tempered reward functions, and using policy noise and weight decay to temper the critic network $Q$. This being said, it would be interesting to build on this work to address other questions.

In the previous section, we established theoretical results to explain how current techniques and practices improve the performance of certain actor-critic methods. An obvious question to address in future work is how such findings could be used for practical algorithm design. One line of thinking might be to develop algorithms that explicitly learn the gradient $\partial Q^\mu / \partial a$, instead of relying on critics to estimate $Q^\mu$ only. Another relevant direction would be the design of methods that specifically temper the critic $Q$ during training. For the sake of illustration, we include a brief example in Appendix C that relates the use of twin critics to the smoothing of the estimator $Q$.

On the theoretical side, the scope of the present article has been quite confined, for both mathematical convenience and readability purposes. On the one hand, we have focused on one aspect of the learning

process, namely the information about gradient error that is contained in the errors $e^B$ and $e^{\Delta Q}$ computed along the paths of the current policy. A natural, though challenging, follow-up question is how much more information about $\partial Q^\mu / \partial a$ can be recovered as the policy evolves and $e^B$ and $e^{\Delta Q}$ are computed along an ever-changing set of trajectories.

We have also restricted our attention to deterministic dynamics in this paper because of our own research interests and because of their relevance to the usual benchmarks for DPG-type algorithms. Extending the discussion to stochastic systems would obviously be relevant for applications related to stochastic optimal control, but would require heavier mathematical machinery.

Of course, many other open questions remain regarding the mathematical basis of actor-critic methods, including convergence conditions (Williams & Baird, 1990), and the effects of sparse training data (Fujimoto et al., 2022).

### Acknowledgments

This work was supported by Mitacs through the Mitacs Accelerate Program.

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

## A  The gradient error and the difference error

Continuing the discussion of section 3.1, the purpose of this appendix is to clarify the link between the difference error

$$e_t^{\Delta Q} \triangleq (Q_{t+1} - Q_t) - (Q_{t+1}^\mu - Q_t^\mu) = \Delta e_t^Q$$

that we study in this paper, and the gradient errors:

$$e_t^{\partial_a Q} \triangleq \frac{\partial Q}{\partial a}(s_t, a_t) - \frac{\partial Q^\mu}{\partial a}(s_t, a_t)$$

and

$$e_t^{\nabla Q} \triangleq \nabla Q(s_{t+1}, \mu(s_{t+1})) - \nabla Q^\mu(s_{t+1}, \mu(s_{t+1})),$$

where as usual, the point $(s_t, a_t) \in \mathcal{S} \times \mathcal{A}$ has an arbitrary action, and $(s_{t+1}, \mu(s_{t+1})) \in \mathcal{S} \times \mathcal{A}$ is the next point along on-policy trajectory following $\mu$, with $s_{t+1} = f(s_t)$. For simplicity, we assume here that the state space and the action space are both Euclidean, specifically that $\mathcal{S} = \mathbb{R}^d$ and $\mathcal{A} = \mathbb{R}^q$.

Defining $\Delta x_t = (s_{t+1} - s_t, \mu(s_{t+1}) - a_t)$, we have the first-order Taylor expansion

$$Q_t = Q_{t+1} - \Delta x_t^\intercal \cdot \nabla Q(s_{t+1}, \mu(s_{t+1})) + \mathrm{O}\left(\|\Delta x_t\|^2\right)$$

for the approximator $Q$, as well as a similar expression for the state-action value function $Q^\mu$. Assuming that $\|\Delta x_t\|^2$ is small enough, we can write that

$$e_t^{\Delta Q} \approx \Delta x_t^\intercal \cdot \left(\nabla Q(s_{t+1}, \mu(s_{t+1})) - \nabla Q^\mu(s_{t+1}, \mu(s_{t+1}))\right)$$
$$= \Delta x_t^\intercal \cdot e_{t+1}^{\nabla Q} = \Delta x_t^\intercal \cdot \left(e_{t+1}^{\partial_a Q}, e_{t+1}^{\partial_s Q}\right), \tag{A.1}$$

where $e_{t+1}^{\partial_s Q}$ is the *state*-gradient error. This equation shows that $e_t^{\Delta Q}$ is *approximately linearly related to the component of $e^{\nabla Q}$ in the direction $\Delta x_t$ at the point $(s_{t+1}, \mu(s_{t+1})) \in \mathcal{S} \times \mathcal{A}$*. On its own, therefore, $e_t^{\Delta Q}$ reflects only the component of $e^{\nabla Q}$ along that single vector $\Delta x_t$.

We can however say more if we consider the additional information provided by a replay buffer. To make our analysis tractable and derive a general result, we will assume that we have an idealized replay buffer, in the sense that it is large enough that all the objects defined below make sense and all the sets are non-empty.

Suppose we have a replay buffer $\mathcal{B} = \{(s_k, a_k, r_k, s'_k)\}_{k=1}^{N_\mathcal{B}}$ where (as is usual in practice) the transitions $(s_k, a_k, r_k, s'_k)$ are not necessarily obtained by following the policy $\mu$ only. Fixing a state $s \in \mathcal{S}$, consider the subset of transition tuples:

$$\mathcal{B}(s) = \left\{ (s_{k_i}, a_{k_i}, r_{k_i}, s'_{k_i}) \mid s'_{k_i} = s, \ i = 1, \cdots, n_s \right\} \subset \mathcal{B}$$

whose next state is $s$. Let us introduce:

$$\Delta x_i \triangleq \left( s'_{k_i} - s_{k_i}, \mu(s'_{k_i}) - a_{k_i} \right),$$
$$e_i^{\Delta Q}(s) \triangleq e^Q\left( s'_{k_i}, \mu(s'_{k_i}) \right) - e^Q(s_{k_i}, a_{k_i}),$$

with $i = 1, \cdots, n_s$. Next, let $e_{\mathcal{B},\mu}^{\Delta Q}(s)$ be the (row) vector whose entries are the $e_i^{\Delta Q}(s)$, let $M_{\mathcal{B},\mu}(s)$ be the matrix whose rows are the vectors $\Delta x_i^\mathsf{T}$, and let $W_{\mathcal{B},\mu}(s) = \mathrm{Span}_\mathbb{R}\{\Delta x_i\}_{i=1}^{n_s}$ be the linear subspace of $\mathcal{S} \times \mathcal{A}$ spanned by the "samples" $\Delta x_i$. By equation (A.1), we have that:

$$e_{\mathcal{B},\mu}^{\Delta Q}(s) \approx M_{\mathcal{B},\mu}(s) \cdot e^{\nabla Q}\left( s, \mu(s) \right), \tag{A.2}$$

which shows that $e_{\mathcal{B},\mu}^{\Delta Q}(s)$ is approximately linearly related to the projection of $e^{\nabla Q}\left( s, \mu(s) \right)$ onto the subspace $W_{\mathcal{B},\mu}(s) \subseteq \mathcal{S} \times \mathcal{A}$. Letting $M_{\mathcal{B},\mu}(s)^+$ denote the Moore-Penrose pseudo-inverse of $M_{\mathcal{B},\mu}(s)$, equation (A.2) tells us that if the $\|\Delta x_i\|$ are small enough, then $M_{\mathcal{B},\mu}(s)^+ \cdot e_{\mathcal{B},\mu}^{\Delta Q}(s)$ is a least-squares estimator for $e^{\nabla Q}\left( s, \mu(s) \right)$.

Given these facts, how much information about $e^{\nabla Q}\left( s, \mu(s) \right)$ or $e^{\partial_a Q}\left( s, \mu(s) \right)$ can be recovered from $e^{\Delta Q}\left( s, \mu(s) \right)$? The answer lies within the subspace $W_{\mathcal{B},\mu}(s)$ spanned by the $\Delta x_i$, and therefore in the samples available in the subset $\mathcal{B}(s)$.

If the examples $\Delta x_i$ are varied enough to span the entirety of $\mathcal{S} \times \mathcal{A}$, then for a given point $s \in \mathcal{S}$, the values $e_i^{\Delta Q}$ will determine all the components of $e^{\nabla Q}\left( s, \mu(s) \right)$, meaning we can completely recover this vector using the difference errors. Similarly, if the $\Delta x_i$ are such that $\mathcal{A} \subseteq W_{\mathcal{B},\mu}(s)$, then the difference errors will allow us to recover the gradient error in the action direction, namely $e^{\partial_a Q}\left( s, \mu(s) \right)$. But these ideal cases are unrealistic in practice, and more generally, it can be shown (though the proofs are beyond the scope of this paper) that even when the space $W_{\mathcal{B},\mu}(s)$ is maximal, in the sense that it contains all *possible* $\Delta x_i$ compatible with the state dynamics and the policy $\mu$, the action space $\mathcal{A}$ will not necessarily be included in $W_{\mathcal{B},\mu}(s)$, and in that case $e^{\Delta Q}$ will not measure how well the critic estimates the projection of $\partial Q^\mu / \partial a$ onto the orthogonal complement $W_{\mathcal{B},\mu}(s)^\perp \subset \mathcal{S} \times \mathcal{A}$.

In summary, the difference error $e^{\Delta Q}$ is not a perfect index of the error in $\partial Q^\mu / \partial a$. In most cases it will contain only incomplete information about $e^{\partial_a Q}$, mixed with inessential information about the state-gradient error $e^{\partial_s Q}$, as shown in A.1. But it is linearly related to the best estimate of $e^{\partial_a Q}$ that can be obtained given the information in the buffer, it is fairly simply related to the Bellman error $e^B$, and it is in any case more informative than the value error $e^Q$ that has been the focus of previous investigations. Our aim in this paper has been to counter the claim that Bellman-based learning is suspect because $e^B$ is a bad surrogate for $e^Q$. To do that, we have shown that $e^{\Delta Q}$ is a more relevant error measure than $e^Q$ in this context, and that $e^B$ is a good surrogate for $e^{\Delta Q}$.

# B   Proof of DFT bound

## B.1   Discrete Fourier Transform

We start with a brief review of the Discrete Fourier Transform (DFT), and establish a lemma that simplifies the proof of Proposition 7.

Recall that given any finite-length real time series $\{x_n\}_{n=0}^{N-1} = \{x_0, x_1, \cdots, x_{N-1}\}$, where $N$ is a positive integer, the DFT of $\{x_n\}_{n=0}^{N-1}$ is the $N$-element sequence $\{\hat{x}_k\}_{k=0}^{N-1}$ whose terms are

$$\hat{x}_k \triangleq \sum_{n=0}^{N-1} x_n \exp(-\mathrm{i}\omega_k n), \quad \forall k = 0, \cdots, (N-1), \tag{B.1}$$

where i is a square root of $-1$, and the frequency variable $\omega_k \triangleq 2\pi k/N$ for all $k = 0, \cdots, (N-1)$.

Given the DFT terms $\{\hat{x}_k\}_{k=0}^{N-1}$, the original time series $\{x_n\}_{n=0}^{N-1}$ is recovered using the identity (Stankovic, 2015, Sec.3.1)

$$x_n = \frac{1}{N} \sum_{k=0}^{N-1} \hat{x}_k \exp(\mathrm{i}\omega_k n), \quad \forall n = 0, \cdots, (N-1),$$

meaning that the quantity $|\hat{x}_k|/N$ represents the amplitude of the component $\exp(\mathrm{i}\omega_k n)$ with frequency $\omega_k$. Moreover, it is a straightforward matter to compute that $\exp(+\mathrm{i}\omega_k) = \exp(-\mathrm{i}\omega_{N-k})$ for $k = 1, \cdots, (N-1)$, which means that $\omega_k$ and $\omega_{N-k}$ represent the same frequency of cycling but in opposite directions — clockwise vs counterclockwise. Therefore the DFT terms $\hat{x}_k$ corresponding to high frequencies are those with $k$ close to $N/2$, while the terms corresponding to low frequencies are those with $k$ close to 0 or $(N-1)$.

Now we show the following general facts:

**Lemma 8.** *Let $\{x_n\}_{n=0}^{N-1}$ be a finite-length discrete time-series.*

1. *The terms of its DFT $\{\hat{x}_k\}_{k=0}^{N-1}$ are given by $\hat{x}_0 = \sum_{n=0}^{N-1} x_n$ and*

$$\hat{x}_k = -\sum_{n=1}^{N-1} \left( \frac{1 - \exp(-\mathrm{i}n\omega_k)}{1 - \exp(-\mathrm{i}\omega_k)} \right) (x_n - x_{n-1}) \tag{B.2}$$

    *for $k = 1, \cdots, (N-1)$.*

2. *For all $k = 1, \cdots, (N-1)$, we have that*

$$|\hat{x}_k| \leq \frac{(N-1)}{\sin\left(\frac{\omega_k}{2}\right)} \Delta x_{\max}, \tag{B.3}$$

    *where $\Delta x_{\max} \triangleq \max_{n=1,\cdots,(N-1)} |x_n - x_{n-1}|$.*

*Proof.* To prove (B.2), we write the terms of the sequence $\{x_n\}_{n=1}^{N-1}$ as a telescoping sum $x_n = x_0 + \sum_{m=1}^{n} (x_m - x_{m-1})$, and using the geometric sums

$$\sum_{n=0}^{m-1} \exp(-\mathrm{i}n\omega_k) = \left( \frac{1 - \exp(-\mathrm{i}m\omega_k)}{1 - \exp(-\mathrm{i}\omega_k)} \right),$$

$$\sum_{n=0}^{N-1} \exp(-\mathrm{i}n\omega_k) = 0,$$

we have from (B.1) that for all $k = 1, \cdots, (N-1)$,

$$\hat{x}_k = \sum_{n=1}^{N-1} \left[ \sum_{m=1}^{n} (x_m - x_{m-1}) \right] \exp(-\mathrm{i}n\omega_k) = \sum_{m=1}^{N-1} \left[ \sum_{n=m}^{N-1} \exp(-\mathrm{i}n\omega_k) \right] (x_m - x_{m-1})$$

$$= -\sum_{m=1}^{N-1} \left[ \sum_{n=0}^{m-1} \exp(-\mathrm{i}n\omega_k) \right] (x_m - x_{m-1}) = -\sum_{m=1}^{N-1} \left( \frac{1 - \exp(-\mathrm{i}m\omega_k)}{1 - \exp(-\mathrm{i}\omega_k)} \right) (x_m - x_{m-1}).$$

And the case of $k = 0$ is trivial.

Next, from the identity $\sin(\theta) = \left[\exp(\mathrm{i}\theta) - \exp(-\mathrm{i}\theta)\right]/2\mathrm{i}$, it follows that for all $m = 0, \cdots, N-1$,

$$\left|\frac{1 - \exp(-\mathrm{i}m\omega_k)}{1 - \exp(-\mathrm{i}\omega_k)}\right| = \left|\frac{2\mathrm{i}\exp\left(-\mathrm{i}m\omega_k/2\right)}{2\mathrm{i}\exp\left(-\mathrm{i}\omega_k/2\right)}\frac{\sin(m\omega_k/2)}{\sin(\omega_k/2)}\right| = \frac{\left|\sin(m\omega_k/2)\right|}{\sin(\omega_k/2)} \leq \frac{1}{\sin(\omega_k/2)}.$$

The right-hand side of equation (B.2) can now be bounded as follows:

$$\left|\hat{x}_k\right| \leq \frac{1}{\sin(\omega_k/2)}\left(\sum_{n=1}^{N-1}\left|x_n - x_{n-1}\right|\right) \leq \frac{(N-1)}{\sin(\omega_k/2)}\max_{n=1,\cdots,(N-1)}\left|x_n - x_{n-1}\right| = \sin\left(\tfrac{\omega_k}{2}\right)^{-1}N\Delta x_{\max},$$

proving equation (B.3). $\qquad\square$

## B.2 Proof of Proposition 7

We now turn to the Bellman error $e_t^B$ and the difference error $e_t^{\Delta Q}$. We have seen that $e_t^{\Delta Q}$ is the result of anticausal high-pass filtering of all the $e_\tau^B$ from $\tau = t$ to $\infty$. But if $e^B$ is upper-bounded by $C > 0$, then for any finite $T$, we have from equation (3.13) that

$$e_t^{\Delta Q} = \sum_{\tau=t}^{T}\gamma^{\tau-t}\left(e_{\tau+1}^B - e_\tau^B\right) + \gamma^{T+1-t}e_{T+1}^{\Delta Q},$$

$$\leq \sum_{\tau=t}^{T}\gamma^{\tau-t}\left(e_{\tau+1}^B - e_\tau^B\right) + 2\gamma^{T+1-t}C,$$

The term $2\gamma^{T+1-t}C$ goes to 0 as $T$ increases, which means that values of $e_\tau^B$ in the distant future have vanishing influence on $e_t^{\Delta Q}$. So with arbitrarily small error, we can consider long but finite time series of Bellman errors, and analyze their frequency components using the DFT terms $\{\hat{e}_k^B\}_{k=0}^{N-1}$. In this context, we derive a refinement of inequality (B.3).

**Lemma 9.** *Suppose that $\{\phi_n\}_{n=0}^{N-1}$ is the discrete time series obtained by evaluating a Lipschitz function $\phi : \mathcal{S} \times \mathcal{A} \to \mathbb{R}$ along a finite-length segment $\{(s_0, a_0),\ (s_n, \mu(s_n))\}_{n=1}^{N-1}$ of a trajectory. The DFT terms $\{\hat{\phi}_k\}_{k=1}^{N-1}$ then satisfy the inequality:*

$$\left|\hat{\phi}_k\right| \leq |\phi_1 - \phi_0| + \frac{(N-2)}{\sin(\omega_k/2)}\mathrm{Lip}(\phi)\sqrt{1 + \mathrm{Lip}(\mu)^2}\left\|f_\mu^\Delta\right\|_{\max}, \tag{B.4}$$

*where $\mathrm{Lip}(\phi)$ and $\mathrm{Lip}(\mu)$ are the Lipschitz constants of $\phi$ and $\mu$ respectively, where $f_\mu^\Delta(s) \triangleq f(s, \mu(s)) - s$, and where $\left\|f_\mu^\Delta\right\|_{\max} = \max_{s \in \mathcal{S}}\left\|f_\mu^\Delta(s)\right\|$.*

*Proof.* Applying equation (B.2) to the DFT terms $\{\hat{\phi}_k\}_{k=1}^{N-1}$ yields:

$$\hat{\phi}_k = -(\phi_1 - \phi_0) - \sum_{n=2}^{N-1}\left(\frac{1 - \exp(-\mathrm{i}n\omega_k)}{1 - \exp(-\mathrm{i}\omega_k)}\right)(\phi_n - \phi_{n-1}),$$

then modifying the proof of (B.3) gives the inequality:

$$\left|\hat{\phi}_k\right| \leq |\phi_1 - \phi_0| + \frac{(N-2)}{\sin(\omega_k/2)}\max_{1 < n \leq N-1}\left|\phi_n - \phi_{n-1}\right|.$$

Since $s_n = f(s_{n-1}, \mu(s_{n-1}))$ for $n = 2, \cdots, N-1$, we have

$$
\begin{aligned}
\left|\phi_n - \phi_{n-1}\right| &= \left|\phi\left(f\left(s_{n-1}, \mu(s_{n-1})\right), \mu\left(f\left(s_{n-1}, \mu(s_{n-1})\right)\right)\right) - \phi\left(s_{n-1}, \mu(s_{n-1})\right)\right| \\
&\leq \mathrm{Lip}(\phi) \cdot \left\|\left(f\left(s_{n-1}, \mu(s_{n-1})\right), \mu\left(f\left(s_{n-1}, \mu(s_{n-1})\right)\right)\right) - \left(s_{n-1}, \mu(s_{n-1})\right)\right\| \\
&\leq \mathrm{Lip}(\phi) \cdot \sqrt{\left\|f\left(s_{n-1}, \mu(s_{n-1})\right) - s_{n-1}\right\|^2 + \mathrm{Lip}(\mu)^2 \left\|f\left(s_{n-1}, \mu(s_{n-1})\right) - s_{n-1}\right\|^2} \\
&\leq \mathrm{Lip}(\phi)\sqrt{1 + \mathrm{Lip}(\mu)^2} \cdot \max_{1 < n \leq N-1} \left\|f\left(s_{n-1}, \mu(s_{n-1})\right) - s_{n-1}\right\| \\
&\leq \mathrm{Lip}(\phi)\sqrt{1 + \mathrm{Lip}(\mu)^2}\left\|f_\mu^\Delta\right\|_{\max}.
\end{aligned}
$$

which completes the proof of the Lemma. $\qquad\square$

Continuing with the notation of this last Lemma, we can now establish:

**Proposition 7.** *Along any finite-length segment $\{(s_0, a_0), (s_n, \mu(s_n))\}_{n=1}^{N-1}$ of a trajectory, the DFT terms $\{\hat{e}_k^B\}_{k=1}^{N-1}$ of the Bellman error satisfy the inequality*

$$
\begin{aligned}
\left|\hat{e}_k^B\right| \leq \frac{(N-2)}{\sin\left(\frac{\omega_k}{2}\right)} &\left\{\|f_\mu^\Delta\|_{\max}\sqrt{1 + \mathrm{Lip}(\mu)^2}\left[(1+\gamma)\mathrm{Lip}(Q) + \mathrm{Lip}(r)\right]\right\} \\
&+ \left[(1+\gamma)\left|Q_1 - Q_0\right| + \left|r_1 - r_0\right|\right] + \gamma\left(|Q_0| + |Q_N|\right).
\end{aligned}
$$

*Proof.* From the definitions $e_n^B = Q_n - r - \gamma Q_{n+1}$ and (B.1), the DFT terms $\hat{e}_k^B$ of the Bellman error are given by

$$
\hat{e}_k^B = \hat{Q}_k - \hat{r}_k - \gamma \sum_{n=0}^{N-1} \exp(-\mathrm{i}n\omega_k)Q_{n+1}, \quad k = 1, \cdots, (N-1),
$$

where $\hat{Q}_k$ and $\hat{r}_k$ are the DFT terms of $Q$ and $r$ respectively. For the sum on the RHS, a direct computation yields

$$
\sum_{n=0}^{N-1} \exp(-\mathrm{i}n\omega_k)Q_{n+1} = \exp(\mathrm{i}\omega_k)\left(\hat{Q}_k - Q_0 + Q_N\right),
$$

where $Q_N = (Q_{N-1} - r_{N-1} - e_{N-1}^B)/\gamma$, so that

$$
\hat{e}_k^B = (1 - \gamma\exp(\mathrm{i}\omega_k))\hat{Q}_k - \hat{r}_k + \gamma\exp(\mathrm{i}\omega_k)(Q_0 - Q_N), \tag{B.5}
$$

and therefore

$$
|\hat{e}_k^B| \leq (1+\gamma)|\hat{Q}_k| + |\hat{r}_k| + \gamma(|Q_0| + |Q_N|).
$$

The inequality in the statement then follows by applying Lemma 9 to the terms $\left|\hat{Q}_k\right|$ and $|\hat{r}_k|$. $\qquad\square$

In summary, the inequality of Proposition 7 relates the modulus of the DFT terms of the Bellman error to the state dynamics and the regularity of the critic $Q$, the reward function $r$, and the policy $\mu$ on the state-action space $\mathcal{S} \times \mathcal{A}$. We also make the following remarks:

(a) Since $|Q_1 - Q_0| = \left|Q\left(s_1, \mu(s_1)\right) - Q(s_0, a_0)\right|$ and $|r_1 - r_0| = \left|r\left(s_1, \mu(s_1)\right) - r(s_0, a_0)\right|$, the term $(1+\gamma)|Q_1 - Q_0| + |r_1 - r_0|$ in the inequality quantifies a gap incurred by transitioning from an arbitrary initial state-action $(s_0, a_0)$ to the on-policy part of the trajectory $\{(s_n, \mu(s_n))\}_{n=1}^{N-1}$. In the case where $a_0 = \mu(s_0)$, the inequality reduces to:

$$
\left|\hat{e}_k^B\right| \leq \frac{(N-1)}{\sin\left(\frac{\omega_k}{2}\right)}\left\{\|f_\mu^\Delta\|_{\max}\sqrt{1 + \mathrm{Lip}(\mu)^2}\left[(1+\gamma)\mathrm{Lip}(Q) + \mathrm{Lip}(r)\right]\right\} + \gamma\left(|Q_0| + |Q_N|\right).
$$

(b) The scale of the term $\|f_\mu^\Delta\|_{\max} = \max_{s \in \mathcal{S}} \|f(s, \mu(s)) - s\|$ depends on the smoothness of the trajectory obtained by following the policy $\mu : \mathcal{S} \to \mathcal{A}$.

(c) To clarify what the term $\|f_\mu^\Delta\|_{\max}$ represents, it might be useful to look at an alternative definition of the state dynamics. Had we adopted the continuous-time dynamical systems notation, our state dynamics would be given by $s_{t+1} = s_t + F(s_t, a_t)$ for a certain function $F : \mathcal{S} \times \mathcal{A} \to \mathcal{S}$. The constant $\|f_\mu^\Delta\|_{\max}$ would then be replaced by $\sup_{s \in \mathcal{S}} \|F(s, \mu(s))\|$.

(d) The bounds on $\left|\hat{e}_k^B\right|$ are much larger for the low frequencies $2\pi k/N$ than for the high frequencies, because $\sin(\omega_k/2)^{-1} \simeq N/\pi k$ for $k$ close to 0 or $(N-1)$ (low frequency), while $\sin(\omega_k/2)^{-1} \simeq 1$ for $k$ close to $N/2$ (high frequency).

## C  On variants of TD3

In this appendix, we briefly describe a possible implication of our results, which may be worth exploring further. Many actor-critic algorithms, starting with TD3 (Fujimoto et al., 2018), use *two* critics (Haarnoja et al., 2018; Wang et al., 2020). Each of the two critics, $Q_i$ for $i = 1, 2$, has its own target, but is trained based on both targets:

$$e_i = Q_i(s, a) - r(s, a) - \gamma \min_{i=1,2} Q_i^{\text{tgt}}(s', \mu(s') + \nu) \tag{C.1}$$

Fujimoto and colleagues advocated taking the minimum of the two target values, as shown in (C.1), on the grounds that critic outputs tend to rise in the course of learning, and so taking the minimum would help prevent $Q$ drifting up and away from the true $Q^\mu$. But our results suggest that there may be value in using the twin targets to *temper* the $Q$-function, so as to reduce its high-frequency components, rather than or in addition to lowering it.

Figure C.1 illustrates this idea. The blue lines in the plots are learning curves for TD3 on two benchmark continuous-control tasks from OpenAI Gym: Ant-v4 and HalfCheetah-v4. The orange lines are learning curves for a very slightly different algorithm, just like TD3 and with the same initializations but training the $Q_i$ based not on the *lower* of the two targets but on the one closer to zero, i.e. on the one whose absolute value is smaller. And the green lines are learning curves for a third version of TD3, where the $Q_i$ are trained based on the *mean* of the two targets. These latter two methods will tend to squeeze the $Q$ function, not just from above as TD3 does, but also from below. The plots indicate that these *small-target* and *mean-target* methods may outperform standard TD3 on some tasks. So there may be value in investigating new ways of handling twin targets that temper $Q$ rather than just lower it.

For each of the two tasks, we ran five trials of each of the three algorithms, all with the same five random seeds, and using the hyperparameters from the TD3 implementation at https://github.com/sfujim/TD3). Of course these results are by no means a thorough test of any algorithm. They are merely a brief illustration of how concepts from this paper might be applied to algorithm design in the future.

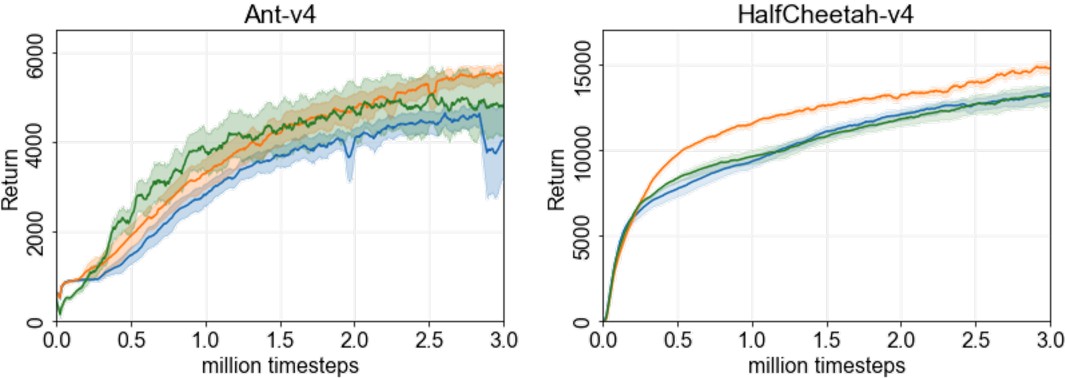

Figure C.1: Learning curves for the standard TD3 algorithm (blue) and two variants of it: a small-target version (orange) that trains its critics based on the smaller of two target-network outputs, and a mean-target version (green) that trains based on the mean of the outputs of the target networks. The shaded regions represent the standard errors of the mean.

