# OpenReview forum: "Error bounds and dynamics of bootstrapping in actor-critic reinforcement learning"
_TMLR — Accepted by TMLR_

### Review · Reviewer_PJ8w · 2023-08-28

**Summary Of Contributions:**

By the deterministic policy gradient theorem, it is important that the gradient of the estimate of the action-value function is accurate. This paper starts from this fact and then makes the following contributions:

This paper introduces the difference error, which better estimates the error in the gradient of the action-value function compared to the value error, and relates the difference error to both the Bellman error and the error in the Q-function estimates.

This paper shows that the difference error and the error in the Q-function estimates are complementary filters of the Bellman error.

This paper establishes a bound on the discrete Fourier coefficients of the Bellman error, which explains how the dynamics of bootstrapping are governed by the regularity of the state dynamics, the policy, the rewards, and the Q-function estimator.

**Audience:**

Yes

**Claims And Evidence:**

No

**Requested Changes:**

Why did you draw $t \to Q(s_t, \mu(s_t))$  instead of $e_t^B$ in Figure 3.1? Don't you want to illustrate Proposition 7, which concerns the DFT terms of $e_t^B$?

In Figure 3.1's caption, you said "In panel(A), the policy is a simple linear one, and the reward function is a simple quadratic with a small Lip(r). Consequently, Lip(Qµ) is also small, and so is Lip(Q), because the critic’s estimate is accurate." If your critic's estimate is accurate, the Bellman error is 0, regardless of the structure of the reward setting and the policy $\mu$. Why did you require that the reward is quadratic and the policy is linear?

Again in Figure 3.1's caption, you said "In panel(B), the critic is less accurate and so Lip(Q) is increased, leading to high-frequency ripples in the time plot." I don't see how the accuracy of the critic is related to Lip Q. What if you have a Q being a constant vector?

In section 2, you need to define the state and action spaces, the reward function.

You need to justify why to consider deterministic environments and policies.

If you have infinite state and action spaces, the series in equation 2.1 may not exist.

Actor-critic methods do not necessarily use a critic "network".

Right above equation 2.2, you need to say what policy $\mu$ is.

Equation 2.7, needs to write $\frac{\partial Q^\mu(s, a)}{\partial a}$. $\frac{\partial Q^\mu}{\partial a}$ does not make sense.

**Strengths And Weaknesses:**

Strengths:
1. Overall the paper is well-written and is easy to understand. Good job!
2. The problem this paper studies is very valuable and interesting.
3. The approach that uses the Fourier analysis and high/low pass filters seems novel to me.

Weaknesses:

1. The main weakness is that it is unclear how the difference error ($(Q(s_{t+1}, a_{t+1}) - Q(s_t, a_t) )- (Q_\mu(s_{t+1}, a_{t+1}) - Q_\mu(s_t, a_t) )$) is related to the partial derivative of the action-value error w.r.t. the action ($\frac{\partial Q(s_t, a_t)}{\partial a_t} - \frac{\partial Q_\mu (s_t, a_t)}{\partial a_t}$). This is a key step that justifies the value of studying the difference error.

2. I am not sure what can I learn from Proposition 7. This proposition shows that both the high-frequency and the low-frequency parts of $e_t^B$ have the same dependence on Lip(\mu), Lip(Q), and Lip(r). Before you said, "The practical consequence is that, for any given magnitude of Bellman errors, the lower we can make the temporal frequency of variation of $e^B_t$, the smaller $e^{\Delta Q}_t$ will be." So from this proposition, I think what we can learn is that by reducing Lip(\mu), Lip(Q), and Lip(r), the magnitude of both high- and low-frequency parts of $e^B_t$ will be smaller. Is this what readers should learn from this proposition? If it is, you should make this point clear.

3. If my previous understanding is correct, I doubt if proposition 7 provides useful knowledge for algorithm design. Specifically, you said "So the lesson of Proposition 7 and Figure 3.1 is that we can shrink the high-frequency components of $e^B_t$ by reducing ||f^{\Delta}_\mu|| or by choosing a simple, low-Lipschitz reward function r or by smoothing out \mu or Q, for instance with weight decay." Both ||f^{\Delta}_\mu||, and r are determined by the problem. And they being low Lipschitz basically means that they don't vary across different state actions. Smoothing out $\mu$ and $Q$ drives them far away from optimal policies and $Q_\mu$. So I don't see why proposition 7 says anything useful about designing algorithms that achieve a low Bellman error.

4. It appears that filters are more like providing a different way to look at the problem, rather than giving new results. For example, equation 3.9, which does not need filters, is enough to show that $e_t^Q$ can be much larger than $e_t^B$. If my understanding is correct, the authors should make this explicit. Right now, it is not clear why you want to introduce filters.

---

> ### Author Response · Authors · 2023-10-04
> **Response to Weaknesses (Reviewer PJ8w)**
>
> Comment 1: The main weakness is that it is unclear how the
> difference error is related to the partial derivative of the action-value
> error w.rt. the action ($\frac{\partial Q(s_{t},a_{t})}{\partial a_{t}}-\frac{\partial Q_{\mu}(s_{t},a_{t})}{\partial a_{t}}$)
>
> Answer: We have rewritten Section 3.1 and added a new Appendix A to address
> this question, as mentioned in item (B) above.
>
> Comment 2: I am not sure what can I learn from Proposition
> 7. This proposition shows that both the high-frequency and the low-frequency
> parts of $e^{B}$ have the same dependence on Lip(\textbackslash mu),
> Lip(Q), and Lip(r). Before you said, \textquotedbl The practical
> consequence is that, for any given magnitude of Bellman errors, the
> lower we can make the temporal frequency of variation of $e_{t}^{B}$,
> the smaller $e_{t}^{\Delta Q}$ will be.\textquotedbl{} So from this
> proposition, I think what we can learn is that by reducing Lip(\textbackslash mu),
> Lip(Q), and Lip(r), the magnitude of both high- and low-frequency
> parts of $e_{t}^{B}$ will be smaller. Is this what readers should
> learn from this proposition? If it is, you should make this point
> clear.
>
> Answer: We have added a new paragraph after Proposition 7, to tie together
> all the results up to that point. This paragraph highlights the fact
> that the high- and low-frequency parts of $e^{B}$ do not have quite
> the same dependence on the Lipschitz factors. Rather, the term $1/\sin(\pi k/N)$
> means that the bounds are much narrower for the high-frequency components
> (those with $k$ near $N/2$). But the paragraph also makes clear
> that reducing the Lipschitz constants affects all frequency components.
>
> Comment 3: I doubt if Proposition 7 provides useful knowledge
> for algorithm design. Specifically, you said "So the
> lesson of Proposition 7 and Figure 3.1 is that we can shrink the high-frequency
> components of $e^{B}$ by reducing $||f_{\mu}^{\Delta}||$ or by choosing
> a simple, low-Lipschitz reward function r or by smoothing out $\mu$
> or $Q$, for instance with weight decay." Both $||f_{\mu}^{\Delta}||$,
> and $r$ are determined by the problem. And they being low Lipschitz
> basically means that they don't vary across different state actions.
> Smoothing out $\mu$ and $Q$ drives them far away from optimal policies
> and $Q_{\mu}$ So I don't see why proposition 7 says anything useful
> about designing algorithms that achieve a low Bellman error.
>
> Answer: To answer these points in order,
> - It is true that the function $f$ defining the system dynamics
> cannot be modified, but the agent may be able to begin learning in
> a simplified environment with smoother dynamics, as in the kiddie-slope
> example at the end of Section 3.3. And we are free to modify the reward
> $r$ to temper its fluctuations. Choosing a good reward function for
> a task is of course a crucial part of reinforcement learning. For
> instance, the analytically correct reward function for time-optimal
> tasks is discontinuous, and using it yields poor results, whereas
> a tempered, continuous approximation to it works much better.
> - The maximal value of $||f_{\mu}^{\Delta}||$ still depends
> on the policy $\mu$.
> - Just to clarify, low Lipschitz means the function doesn't change
> too abruptly, not that it ``doesn't vary across state-action space''.
> A function may vary a lot across a wide domain but still have a low
> Lipschitz constant.
> - We agree that smoothing out $\mu$ and $Q$ could drive them away
> from optimal policies and from $Q^{\mu}$, but the same objection
> can be raised against almost any form of regularization. For instance
> weight decay simply drags all weights toward 0, dropout interferes
> with accurate gradient descent, and policy noise perturbs the Bellman
> error away from its true value, as in our equation (3.14). And all
> these methods do indeed disrupt learning if they are applied too vigorously.
> In each case, empirical tests have been required to establish the
> hyperparameters that make the methods useful.
>
> Comment 4: It appears that filters are more like providing
> a different way to look at the problem, rather than giving new results.
> For example, equation 3.9, which does not need filters, is enough
> to show that $e_{t}^{Q}$ can be much larger than $e_{t}^{B}$. If
> my understanding is correct, the authors should make this explicit.
> Right now, it is not clear why you want to introduce filters.}
>
> Answer: In our view, equation 3.9 (numbered 3.10 in the new version) is one
> way of expressing the fact that $e_{t}^{Q}$ is a low-pass filter.
> It is true that we could have derived that equation without mentioning
> filters, but identifying $e_{t}^{Q}$ and $e_{t}^{\Delta Q}$ explicitly
> as filters provides a valuable framework for understanding them. For
> instance, it was our realization that $e_{t}^{Q}$ is a filter that
> led us to equation 3.9 (now 3.10), which is simply the expression
> of the filter as a convolution. Similarly, filter concepts gave us
> a short proof of Corollary 6, and led us to consider the DFT of $e^{B}$.

---

> ### Author Response · Authors · 2023-10-04
> **Response to Requested Changes (Reviewer PJ8w)**
>
> 1) Why did you draw $t\mapsto Q\left(s_{t},\mu(s_{t})\right)$
> instead of $e_{t}^{B}$ in Figure 3.1? Don't you want to illustrate
> Proposition 7, which concerns the DFT terms of $e_{t}^{B}$?
>
> Ans: We have added several paragraphs to Section 3.3 to explain Figure
> 3.1 more clearly. We plotted $t\mapsto Q(s_{t},\mu(s_{t}))$ instead
> of $e_{t}^{B}$ for several reasons:
> - The 3D graph of the function $e^{B}:\mathcal{S}\times\mathcal{A}\to\mathbb{R}$
> is more difficult to interpret than that of $Q$ .
> - The terms that contribute most to the high-frequency $|\hat{e_k^{B}}|$
> come from the DFT terms of $t\mapsto Q\left(s_{t},\mu(s_{t})\right)$ and of $t\mapsto r\left(s_{t},\mu(s_{t})\right)$.
> - A subtlety that is hard to depict with $e_{t}^{B}$ is how
> $\mathrm{Lip}(Q)$ is affected by $\mathrm{Lip}(r)$ and $\mathrm{Lip}(\mu)$.
>
> The plots in this figure look different from those in the original
> manuscript because we altered the simulations to use the same common
> value of the discount factor, $\gamma$ = 0.99, that we refer to elsewhere
> in the text, and to yield simple, small numbers for the axis labels,
> so as not to clutter the figure.
>
> 2) In Figure 3.1's caption, you said "In
> panel(A), the policy is a simple linear one, and the reward function
> is a simple quadratic with a small Lip(r). Consequently, Lip(Qu) is
> also small, and so is Lip(Q), because the critic's
> estimate is accurate." If your critic's estimate is
> accurate, the Bellman error is 0, regardless of the structure of the
> reward setting and the policy $\mu$. Why did you require that the
> reward is quadratic and the policy is linear?
>
> Ans: Figure 3.1A makes the point that spatially-smooth $f$, $r$, $\mu$,
> and $Q$ result in a temporally smooth $Q_{t}$. But the $Q$ surface
> in 3.1A is not just smooth, it is also a very good approximation to
> $Q^{\mu}$, and that is a nice feature because it means that panels
> C and D can then show how $r$ and $\mu$ affect $Q$ indirectly,
> by roughening $Q^{\mu}$ itself. We could not have illustrated these
> effects using plots of $e^{B}$ because, as you say, those plots would
> have been flat whenever $Q$ was accurate.
>
> Regarding the linear policy and the quadratic reward, these choices
> were made for their simplicity, and also for their physical significance
> (the policy is a force and the reward involves the kinetic energy)
> and for the fact that they led to well-behaved trajectories and state-action
> value function.
>
> 3) Again in Figure 3.1's caption, you said "In
> panel(B), the critic is less accurate and so Lip(Q) is increased,
> leading to high-frequency ripples in the time plot."
> I don't see how the accuracy of the critic is related to $\mathrm{Lip}(Q)$.
> What if you have a $Q$ being a constant vector?
>
> Ans: We did not mean to imply that accuracy always implies a low $\mathrm{Lip}(Q)$.
> It is of course easy to find examples where $Q^{\mu}$ is bumpy
> and $Q$ is a poor approximation but happens to very smooth or even
> constant, as you say. Therefore in the caption we have now replaced
> "less accurate'' with "coarse approximator'' to clarify our
> point -- that $Q$ had 500 Gaussian filters in panel A, but only
> 50 in panel B, and so in B it managed only a coarse approximation
> to the smooth $Q^{\mu}$, with more abrupt variations locally, which
> increased $\mathrm{Lip}(Q)$.
>
> 4) In section 2, you need to define the state and
> action spaces, the reward function.
>
> Ans: We have added two sentences in the first paragraph of section 2, one
> to specify that $\mathcal{S}\subseteq\mathbb{R}^{d}$ and $\mathcal{A}\subseteq\mathbb{R}^{p}$,
> and another one to clarify that $r:\mathcal{S}\times\mathcal{A}\to\mathbb{R}$.
> We prefer to keep $\mathcal{S}$ and $\mathcal{A}$ as general as
> possible for the sake of readability and to avoid cluttering with
> technical adjectives.
>
> 5) You need to justify why to consider deterministic
> environments and policies.
>
> Ans: We have rewritten the end of Section 2.1 to justify our restriction
> to deterministic environments and off-policy DPG-based algorithms.
>
> 6) If you have infinite state and action spaces,
> the series in equation 2.1 may not exist. }
>
> Ans: We now specify in the first paragraph of Section 2 that we are assuming
> the boundedness of $r:\mathcal{S}\times\mathcal{A}\to\mathbb{R}$,
> to ensure that the sum in eq. (2.1) converges.
>
> 7) Actor-critic methods do not necessarily use a
> critic "network".
>
> Ans: We have reworded paragraphs 2 and 4 of section 2 to include more general
> function approximators.
>
> 8) Right above equation 2.2, you need to say what
> policy $\mu$ is.
>
> Ans: The first paragraph of section 2 now specifies the domain and range
> of the policy.
>
> 9) Equation 2.7, needs to write $\frac{\partial Q^{\mu}(s,a)}{\partial a}$.
> $\frac{\partial Q^{\mu}}{\partial a}$ does not make sense.
>
> Ans: We have modified equation (2.7).

---

### Review · Reviewer_4gdd · 2023-09-04

**Summary Of Contributions:**

The paper argues that minimizing the Bellman error is not the ideal objective for off-policy actor critic methods, because the deterministic policy gradient only requires an approximation of the $\frac{\partial Q(\mathbf{s}, \mathbf{a})}{\partial \mathbf{a}}$, but not necessarily a good approximation of $Q(\mathbf{s},\mathbf{a})$. Instead, the paper proposed to minimize the "difference error" the true difference in Q values along subsequent states, and the corresponding according to the learned Q function. The paper argues that the value-error corresponds to a low-pass filter of the Bellman errors along a trajectory, whereas the proposed difference error corresponds to a high-pass filter. Based on this perspective, the paper evaluates a variation of TD3 that instead of using the minimum of both Q-networks, use the one with smaller absolute value, and shows that this procedure slightly improves learning performance on two selected Mujoco experiments.

**Audience:**

No

**Claims And Evidence:**

No

**Requested Changes:**

* The experimental evaluation needs to be significantly expanded. More different environments need to be considered, ideally with different forms of reward function (going beyond locomotion). The procedure for selecting hyperparameters, and the chosen hyperparameters need to be clearly stated.

* The main insight needs to be clearly stated.

* The difference error needs to be better motivated, in particular the effect of state dependent errors $c(t)$ needs to be better discussed.

**Strengths And Weaknesses:**

Soundness
=========

The motivation of the difference error (Eq. 3.1) is not fully convincing. As we are only interested in the accuracy of the gradient of the Q function *with respect to the action*, the error in our Q values does not need to be absolutely constant ($c$), but it suffices if it is constant in actions ($c(s)$). However, the difference error compares Q-values of two different time steps that often involve different states. Hence, if the learned Q-function differs from the true Q function only in terms of an arbitrary state value error $c(s)$, the difference Error can be large, despite resulting in the correct partial derivative wrt the action.

I do not see the purpose of Proposition 2. Doesn't it just state that the loss function tends to take smaller absolute values?

Corollary 6 seems to be wrong and useless. Consider a very large C, large enough to ensure that it upper-bounds the Bellman error. The corollary states that any Q function with a large value error of $\frac{1}{\gamma}C$ at $(s\_0, a\_0)$ will have zero temporal difference error at every time step. Which clearly can not be the case in general. The proof states that $e\_{t}^{Q}$ cann only be non-zero at $(s\_0, a\_0)$ if the Bellman error is constant. However, this seems to assume that the Q function is trained in a certain way, which is not stated in the Corollary. Furthermore, what is the point of assuming that the Q function is trained along a single infinite trajectory (which is never the case in practice) and that this training will converge to a particular value (without providing any reason why it should converge to that value)?
The next paragraph argues that this leads to a partial tradeoff: When the Bellman error is constant, the Bellman difference error will be zero, and hence, the paper argues, when maximizing the Bellman error, it will be always at the upper bound $C$ and therefore constant. However, I don't think that this argumentation makes sense, because maximizing the Bellman error would in general not make it constant.

The paper argues that Proposition 7 (a bound on the Bellman-error) helps to better understand the benefit of policy noise. Whereas Fujimoto et al (2018) introduced it as a kind of regularization to smoothen the Q-function wrt the actions to reduce overfitting , the paper argues, that it should rather be understood as a method for smoothing the Q-values wrt time. However, in my opinion this motivation is less convincing than the one proposed by Fujimoto because policy noise is only very indirectly related to temporal smoothness.

All the analysis in the paper considers the different errors when assuming a fixed trajectory. It is not clear how this relates to replay buffers consisting of many different episodes and different policies.

Clarity
=======
The purpose of Fig. 3.1 is not clear. It shows different variation of the same simple task and the state-action values along a trajectory. I think the intention of the figure is to show that more difficult task and function approximation errors can lead to less smooth (in time) Q-values. However, four simple examples are not are not indicative of a general trend (I do agree that there should be a general tendencies to smoother Q-functions for simpler tasks, so even if the results were significant, they were hardly surprising). In terms of presentation, one needs to read a rather long paragraph in the caption to understand what the figure is showing. Adding legends would would be preferable.

The function $f$ is not sufficiently well defined. It seems to be refering to the closed-loop deterministic system dynamics for a given policy.

Evaluation
==========
The experimental evaluation is insufficient to undermine the claim that using Q networks with smaller absolute values outperform using the minimum Q values. The paper only shows two Mujoco environments. Furthermore, the paper contains insufficient details about the chosen hyperparameters, and the procedure for selecting them.

---

> ### Author Response · Authors · 2023-10-04
> **Response to Summary of Contributions (Reviewer 4gdd)**
>
> Comment: Based on this perspective, the paper evaluates a
> variation of TD3 that instead of using the minimum of both Q-networks,
> use the one with smaller absolute value, and shows that this procedure
> Slightly improves learning performance on two selected Mujoco experiments.
>
> Ans: There seems to be a confusion on this point, for which we apologize.
> We answer this remark in point (D) of our general response, and we
> have modified Section 4 and added Appendix C accordingly.

---

> ### Author Response · Authors · 2023-10-04
> **Response to Soundness, Part 1 (Reviewer 4gdd)**
>
> 1) The motivation of the difference error (Eq. 3.1)
> is not fully convincing. As we are only interested in the accuracy
> of the gradient of the Q function with respect to the action, the
> error in our Q values does not need to be absolutely constant (c),
> but it suffices if it is constant in actions (c(s)). However, the
> difference error compares Q-values of two different time steps that
> often involve different states. Hence, if the learned Q-function differs
> from the true Q function only in terms of an arbitrary state value
> error c(s), the difference Error can be large, despite resulting in
> the correct partial derivative wrt the action.
>
> Ans: Please refer to point (B) of our response letter, which summarizes
> how our revised Section 3.1 and the new Appendix A address this remark.
>
> 2) I do not see the purpose of Proposition 2. Doesn't
> it just state that the loss function tends to take smaller absolute
> values?
>
> Ans: Proposition 2 states that if $|e_{t}^{B}|\le C$ for all $t$, then
> $|e_{t}^{\Delta Q}|\le2C/\gamma$, unlike $|e_{t}^{Q}|$ which is
> upper-bounded by the larger quantity $C/(1-\gamma)$. The point is
> that while $e_{t}^{B}$ is a bad proxy for $e_{t}^{Q}$, it is a good
> one for $e_{t}^{\Delta Q}$.
>
> 3) Corollary 6 seems to be wrong and useless. Consider
> a very large $C$, large enough to ensure that it upper-bounds the
> Bellman error. The corollary states that any $Q$ function with a
> large value error of $\frac{1}{\gamma}C$ at $(s_{0},a_{0})$ will
> have zero temporal difference error at every time step. Which clearly
> can not be the case in general. The proof states that $e_{t}^{Q}$
> cann only be non-zero at $(s_{0},a_{0})$ if the Bellman error is
> constant. However, this seems to assume that the $Q$ function is
> trained in a certain way, which is not stated in the Corollary. Furthermore,
> what is the point of assuming that the $Q$ function is trained along
> a single infinite trajectory (which is never the case in practice)
> and that this training will converge to a particular value (without
> providing any reason why it should converge to that value)? The next
> paragraph argues that this leads to a partial tradeoff: When the Bellman
> error is constant, the Bellman difference error will be zero, and
> hence, the paper argues, when maximizing the Bellman error, it will
> be always at the upper bound C\textquoteright{} and therefore constant.
> However, I don't think that this argumentation makes sense, because
> maximizing the Bellman error would in general not make it constant.
>
> Ans: There is a lot to unpack here. The corollary states that if $e^{Q}$
> reaches its upper-bound $C/(1-\gamma)$ at $(s_{0},a_{0})$ then the
> \emph{difference error} $e^{\Delta Q}$ vanishes along the trajectory
> starting at $(s_{0},a_{0})$ and following $\mu$. The proof states
> that $e_{t}^{Q}$ can be \emph{maximal} (i.e. equal to $C/(1-\gamma)$)
> at $(s_{0},a_{0})$ only if the Bellman error thereafter is constant,
> not that $e_{t}^{Q}$ can be \emph{non-zero} only under those conditions.
> The paragraph following the proof has now been rewritten to make clear
> that the trade-off concerns $e_{t}^{Q}$ and $e_{t}^{\Delta Q}$;
> it refers to the upper bound on $e_{t}^{B}$ but does not say anything
> about maximizing $e_{t}^{B}$ or making it constant.
>
> The corollary relies on no assumptions about how $Q$ is or has been
> trained, neither about trajectories or convergence. We have added
> a paragraph to the end of Section 2.2 to clarify this point.
>
> Finally, the corollary and its proof are entirely correct. The claim
> is that if $e^{Q}$ hits its maximum possible value at $(s_{0},a_{0})$,
> then indeed $e^{B}=C$ along the entirety of the trajectory starting
> as $(s_{0},a_{0})$ and following $\mu$. To derive this result in
> other way, suppose for the sake of contradiction that there exists
> some $i\ge0$ for which $e_{i}^{B}\ne C$, so that $e_{i}^{B}<C$
> necessarily. Notice then that $C/(1-\gamma)=\sum_{t\ge0}\gamma^{t}C$,
> so that by Proposition 4, the equation $e^{Q}(s_{0},a_{0})=C/(1-\gamma)$
> can be rearranged to:
>
> $$\sum_{t=0}^{\infty}\gamma^{t}(C-e_{t}^{B})=\left[\sum_{t\ne i}\gamma^{t}(C-e_{t}^{B})\right]+\gamma^{i}(C-e_{i}^{B})=0.$$
>
> Since $\gamma^{i}(C-e_{i}^{B})>0$, we must then have $\left[\sum_{t\ne i}\gamma^{t}(C-e_{t}^{B})\right]<0$,
> which is absurd since it is a convergent series of non-negative numbers.

---

> ### Author Response · Authors · 2023-10-04
> **Response to Soundness, Part 2 (Reviewer 4gdd)**
>
> 4) The paper argues that Proposition 7 (a bound on
> the Bellman-error) helps to better understand the benefit of policy
> noise. Whereas Fujimoto et al (2018) introduced it as a kind of regularization
> to smoothen the Q-function wrt the actions to reduce overfitting ,
> the paper argues, that it should rather be understood as a method
> for smoothing the Q-values wrt time. However, in my opinion this motivation
> is less convincing than the one proposed by Fujimoto because policy
> noise is only very indirectly related to temporal smoothness.
>
> Ans: We agree that policy noise regularizes the critic, or in other words
> smooths $Q:\mathcal{S}\times\mathcal{A}\to\mathbb{R}$. Our point
> is that this spatial smoothing will reduce the temporal fluctuations
> of $Q_{t}$ and therefore of $e_{t}^{B}$, which will in turn reduce
> $e_{t}^{\Delta Q}$, owing to its high-pass filter properties. We
> have reworded part of the final paragraph in Section 3.4 to clarify
> this point.
>
> 5) All the analysis in the paper considers the different
> errors when assuming a fixed trajectory. It is not clear how this
> relates to replay buffers consisting of many different episodes and
> different policies.
>
> Ans: We address this question in our new Appendix A.

---

> ### Author Response · Authors · 2023-10-04
> **Response to Clarity, Evaluation, and Requested Changes (Reviewer 4gdd)**
>
> Clarity:
>
> 1) The purpose of Fig. 3.1 is not clear. It shows
> different variation of the same simple task and the state-action values
> along a trajectory. I think the intention of the figure is to show
> that more difficult task and function approximation errors can lead
> to less smooth (in time) Q- values. However, four simple examples
> are not are not indicative of a general trend (I do agree that there
> should be a general tendencies to smoother Q-functions for simpler
> tasks, So even if the results were significant, they were hardly surprising).
> In terms of presentation, one needs to read a rather long paragraph
> in the caption to understand what the figure is showing. Adding legends
> would would be preferable.
>
> Ans: We answer this comment in point (C) of our cover letter.
>
> 2) The function f is not sufficiently well defined.
> It seems to be refering to the closed-loop deterministic system dynamics
> for a given policy.
>
> Ans: $f$ is defined in lines 8 and 9 of the first paragraph of Section
> 2.1.
>
> Evaluation:
>
> 1) The experimental evaluation is insufficient to undermine
> the claim that using Q networks with smaller absolute values outperform
> using the minimum Q values. The paper only shows two Mujoco environments.
> Furthermore, the paper contains insufficient details about the chosen
> hyperparameters, and the procedure for selecting them.
>
> Ans: We address this question in point (D) of our cover letter.
>
> Requested changes:
>
> 1) The experimental evaluation needs to be significantly
> expanded. More different environments need to be considered, ideally
> with different forms of reward function (going beyond locomotion).
> The procedure for selecting hyperparameters, and the chosen hyperparameters
> need to be clearly stated.
>
> Ans: Please refer to point (D) of our cover letter.
>
> 2) The main insight needs to be clearly stated.
>
> Ans: In the revised manuscript, this point is addressed in paragraph 3
> of the introduction.
>
> 3) The difference error needs to be better motivated,
> in particular the effect of state dependent errors c(t) needs to be
> better discussed.
>
> Ans: Please refer to point (B) of our cover letter.

---

> ### Comment · Reviewer_4gdd · 2023-10-09
> **Thank you for the clarifications**
>
> Thank you for the clarifications. However, I still have several open questions:
>
> Based on the revision, I think that the main insight of the submission is supposed to be a better justification for minimizing the Bellman error, which is summarized as follows in the last sentence of Appendix A:
> > To do that, we have shown that $e^{\Delta Q}$ is a more relevant error measure than $e^Q$ in this context, and
> > that $e^B$ is a good surrogate for $e^{\delta{Q}$.
>
> However, I still think that both parts of this sentence are not sufficiently well substantiated. In what sense does the paper prove that  $e^{\Delta Q}$ is more relevant than $e^Q$? I agree that the action-gradient error is more relevant than the Q-error, but why exactly is the difference error more relevant? What does it matter that the Q-error at a single step does not contain information about the gradient, when we fit the Q-function over a larger state-action space? Regarding the second part of the statement: Why exactly is  $e^B$ a good surrogate for $e^{\Delta{Q}}$? Because there exists an upper bound on $e^{\Delta{Q}}$ as a function of the maximum of $e^B$, that happens to be smaller than some other upper bound on $e^{Q}$? Is this a sufficient condition for a "better" proxy?
>
> I also still don't see the usefulness of Corollary 6:
> > In words, if the Fujimoto et al. bound (3.4) is tight at any point in state-action space [...]
>
> Why would it ever be tight at any point in state-action space? What is the relevance of this statement?
>
> > Another practical consequence, more clearly expressed by equation (3.13), is that for any given magnitude of the Bellman error, [..]
>
> When would we have a given magnitude of the Bellman error? In what setting would this be a fixed value? Doesn't it change whenever we change the Q-function, or anything in the MDP?
> `

---

> > ### Author Response · Authors · 2023-10-12
> > **Response to Reviewer 4gdd's questions on 23/10/09 - Part 1**
> >
> > 1) In what sense does the paper prove that $e^\Delta{Q}$ is more relevant than $e^Q$? I agree that the action-gradient error is more relevant than the Q-error, but why exactly is the difference error more relevant?
> >
> > Ans: Agreeing that the action-gradient error is more relevant, the first objective is to relate the bounds on $e^B$ and $e^Q$ to the bound on the gradient error. At this point a major issue arises, because in Bellman-based training of the critic, there is no natural way of separating the action-gradient error from the state-gradient error along a trajectory following a fixed policy. In this setting, we therefore have no way of relating the action-gradient error to $e^B$ or $e^Q$.
> > This leads us to considering the gradient error $e^{\nabla Q}$ discussed in Sec.3.1 and Appendix A, which in turn requires us to introduce an idealized replay-buffer. As explained in Appendix A, the difference error is our best proxy for $e^{\nabla Q}$ and the action-gradient error, and compared to $e^Q$, it is a better measure of the quality of the action-gradient of the critic for the following reasons:
> >
> > i) As we can see from eq.(A.1), a small $|e^{\Delta Q}|$ implies that the projections of $e^{\partial_a Q}$ and $e^{\nabla Q}$ onto the space spanned by the replay-buffer examples are also of small norm.
> >
> > ii) In contrast, $|e^Q|$ can be very large, even when all the components of the action-gradient error or $e^{\nabla Q}$ are very small, and $|e^Q|$ can be minuscule when all the components of the action-gradient error or $e^{\nabla Q}$ are very large. In other words, $e^{\Delta Q}$ may be missing information about some components of the gradient, but $e^Q$ can be misleading about all components.
> >
> > Last but not least, as it turns out from the results of section 3.2, $e^{\Delta Q}$ is closely related to the Bellman error, which is why we center our discussion on it.
> >
> > 2) What does it matter that the Q-error at a single step does not contain information about the gradient, when we fit the Q-function over a larger state-action space?
> >
> > Ans: Assuming that Q has already been trained over several thousand trajectories, the error $|e^Q|$ can still be substantially large, and what we really care about in the end is the action-gradient of Q. This takes us back to point (1) above.
> >
> > 3) Regarding the second part of the statement: Why exactly is $e^B$ a good surrogate for $e^{\Delta Q}$? Because there exists an upper bound on $e^{\Delta Q}$ as a function of the maximum of $e^B$, that happens to be smaller than some other upper bound on $e^Q$? Is this a sufficient condition for a "better" proxy?
> >
> > Ans: If the upper-bound on $|e^B|$ is closer to the upper-bound on $|e^{\Delta Q}|$ than it is to that of $|e^Q|$, by 2 orders of magnitude, we obviously know much more about the scale of $|e^{\Delta Q}|$ than that of $|e^Q|$ once we compute $e^B$. In our view, this is the very definition of "$e^B$ is a better proxy for $e^{\Delta Q}$".

---

> > ### Author Response · Authors · 2023-10-12
> > **Response to Reviewer 4gdd's questions on 23/10/09 - Part 2**
> >
> > 4) I also still don't see the usefulness of Corollary 6. Why would it ever be tight at any point in state-action space? What is the relevance of this statement?
> >
> > Ans: Once we know how the bounds on $e^Q$ and $e^{\Delta Q}$ are related to the bound on $e^B$, a natural question to ask is "what happens to $e^{\Delta Q}$ in the worst case scenario where $|e^Q|$ is maximal?", because à priori, nothing prevents $e^Q$ from reaching its maximal possible value over state-action space. This is the first reason why this result is relevant. The second reason why this result is relevant is precisely its statement: If $|e^Q(s,a)|$ is maximal, then $|e^{\Delta Q}|=0$ along the $\mu$-trajectory starting at $(s,a)$. This statement should be deeply disconcerting, because it is saying that if $Q$ is as bad as it could get at a given point $(s,a)$, its derivative along the trajectory is precisely that of the true action-value function $Q^{\mu}$. The next thing to do then is to investigate how one can lower $|e^{\Delta Q}|$ by controlling the functions that produce $e^B$, which leads us to the results on controlling the temporal frequency.
> >
> > 5) When would we have a given magnitude of the Bellman error? In what setting would this be a fixed value? Doesn't it change whenever we change the Q-function, or anything in the MDP?
> >
> > Ans: In the deterministic environments that we are mostly interested in, the state and action spaces are closed and bounded. On the other hand, the rewards, Q-function and policy $\mu$ are all Lipschitz continuous. Under these circumstances, we are forcing $e^B$ to be bounded.
> > When we update the weights and biases of either $\mu$ or $Q$, we are still presumably working with Lipschitz continuous functions, meaning that we still have a bounded $e^B$, which still controls the new upper bound on $|e^{\Delta Q}|$. The same is true if we change the reward to another Lipschitz continuous function.
> >
> > At this point, it is perhaps useful to take a step back and look at the big picture of this paper. In 2022, Fujimoto and colleagues wrote a paper called “Why should I trust you, Bellman? The Bellman error is a poor replacement for value error”. In that paper, they derived a bound showing that $|e^B|$ can be very small while $|e^Q|$ can be very large. Based on this fact, Fujimoto et al. ask the question: "why should we trust that learning based on the Bellman error will lead to a small value error when the two errors are so loosely related?"; and argue that the discrepancies between $e^B$ and $e^Q$ undermines the rationale for Bellman learning.
> >
> > In our work, we show that the loose relation between Bellman error and value error is not a reason to mistrust Bellman, because a small value error is not necessary for good policy training. What is important is having good quality action-gradients for the critic, and we show how this is related to the Bellman error through the difference error. All the modifications we made in our manuscript were made to clarify this narrative.
> >
> > We hope these responses clarify our argument. If there are further concerns, we may be able to answer them more effectively if we understand better the reviewer's standpoint: is the reviewer saying that $e^Q$ is actually an excellent measure, and therefore Fujimoto's analysis is correct and Bellman-learning is unjustified; or does the reviewer agree with us that Bellman can be justified by considering another error measure, but thinks that our analysis relating $e^B$ to $e^{\Delta Q}$ and $e^{\Delta Q}$ to $e^{\partial_a Q}$ is irrelevant to that question?

---

### Review · Reviewer_KAiw · 2023-09-21

**Summary Of Contributions:**

The paper studies the use of Bellman Error (BE)-minimizing algorithms for learning the critic in deterministic actor-critic (AC) algorithms. Specifically, it sheds light on why BE-based critics perform well in DDPG, TD3, and SAC when they tend to not be relied on by other AC algorithms like PPO due to their high Value Error (VE). The paper shows that this is because the deterministic AC algorithms only use the critic to estimate the direction to improve the policy, a shift-invariant quantity that is not harmed as badly by the BE being a poor proxy of VE than AC algorithms that rely on critic estimates to have low VE.

Specifically, the paper:
1. points out that deterministic AC algorithms only use the critic to estimate the direction of policy improvement (undervalued insight in my opinion).
2. proposes the Difference Error as a more relevant measure of error for this purpose.
3. derives a bound on the Value Error in terms of the Difference Error, and shows that it's significantly better than the corresponding bound in terms of the Bellman Error.
4. interprets the VE as a low-pass filter of the BE and Difference Error as a high-pass filter of the BE, to show why the bounds are so different and why the VE can grow so much larger than the BE.
5. uses the discrete Fourier transform to break the Bellman Error of a trajectory into frequency terms and bounds their magnitude in terms of several quantities.
6. discusses possible approaches to reducing these terms, and demonstrates that two specific approaches improve performance on Ant-v4 and HalfCheetah-v4.

**Audience:**

Yes

**Broader Impact Concerns:**

I don't see any ethical issues specific to this paper beyond those faced by the whole field.

**Claims And Evidence:**

Yes

**Requested Changes:**

### Important changes:
- More information about the plot could be helpful for understanding it. For example, the plot of the state space is 2D, so presumably the position and velocity of the bead on the wire? All axis information would be good to include on the plots, actually.
- More information about the experimental methodology for the plots in Figure 4.1. What hyperparameters were used for each method? Are the shaded regions confidence intervals? If so, what confidence level?
- It would be much better to present evidence that the performance gain observed in the demonstrations is due to the changes made in the algorithms. Consider building on Section 3.3 by comparing the magnitudes of the DFT terms of the observed Bellman Errors for each of the algorithms. If the modified algorithms exhibit smaller high-frequency components than the unmodified algorithm, that would be much more convincing than simply showing a performance improvement.

### Less important changes:
#### Section 2.1
- The reward and state transition functions are not usually defined as deterministic. This would be a big change from the usual setting, and if the analysis relies on this fact it could limit the usefulness of the paper's findings (a lot of issues in RL go away with deterministic rewards and transitions).
- Similarly, the agent's policy is not necessarily (or even usually) deterministic. Do the authors mean that the policy is deterministic in the algorithms considered in the paper?
- The discount factor can be 0 (then value becomes immediate reward), but not 1.
- Much of the presentation in this section is specific to a subset of RL, and it would be better to be clear about this. E.g., $Q^\mu(s,\mu(s)) = V^\mu(s)$ is only true for deterministic policies.
- It would be good to more clearly state that the Off-Policy Deterministic Policy Gradient theorem these algorithms rely on is based on an approximation (dropping the gradient of the action value function from the derivation entirely) that is only valid in the tabular setting (see [Errata B in Degris et al. (2012)](https://arxiv.org/pdf/1205.4839.pdf)), and can be a problem when using the resulting algorithms with function approximation ([Imani, 2018](https://proceedings.neurips.cc/paper_files/paper/2018/file/3ef815416f775098fe977004015c6193-Paper.pdf)).

#### Section 3.1:
- "This bound helps explain why DPG-based actor-critic learning works well in practice." It might be better to explicitly say why, as this was not clear to me at first. Is it because DPG-based AC algorithms use the BE-minimizing critic to estimate partial derivatives (that are shift-invariant), which is harmed less by the BE being a bad proxy for value error than algorithms that use the critic in ways that depend on low VE?

#### Section 3.2:
- More context/explicit explanation would be helpful for the last paragraph of 3.2. Why would we want $e^{\Delta Q}_t$ to be small? I can infer it's because a smaller difference error means better estimates of partial derivatives which means better actor updates?

**Strengths And Weaknesses:**

### Strengths:
- The paper is written clearly and concisely, and was very easy to understand. Thank you!
- Proofs are written in-line, presented simply, and explained very clearly. Again, thank you, authors.
- Sheds light on a problem that's very confusing (why deterministic AC algorithms are able to perform well with terrible critics) and under-studied.

### Weaknesses:
- The paper doesn't validate their theory with extensive empirical results, but does provide a demonstration. This isn't a dealbreaker for me, but the paper would be a lot stronger with more extensive experiments/demonstrations.
- Details on plots and experiments are missing.

---

> ### Author Response · Authors · 2023-10-04
> **Response to Important Changes (Reviewer KAiw)**
>
> 1) More information about the plot could be helpful
> for understanding it. For example, the plot of the state space is
> 2D, so presumably the position and velocity of the bead on the wire?
> All axis information would be good to include on the plots, actually.
>
> Ans: We address this question in point (C) of our response letter. We now
> label all the plot axes, and in the text we specify the state space,
> policies, and rewards.
>
> 2) More information about the experimental methodology
> for the plots in Figure 4.1. What hyperparameters were used for each
> method? Are the shaded regions confidence intervals? If so, what confidence
> level?
>
> Ans: We now specify the hyperparameters in the main text, and in the figure
> caption we say that the shaded regions represent the standard errors
> of the mean.
>
> 3) It would be much better to present evidence that
> the performance gain observed in the demonstrations is due to the
> changes made in the algorithms. Consider building on Section 3.3 by
> comparing the magnitudes of the DFT terms of the observed Bellman
> Errors for each of the algorithms. If the modified algorithms exhibit
> smaller high-frequency components than the unmodified algorithm, that
> would be much more convincing than simply showing a performance improvement.
>
> Ans: Thank you for the suggestion. We address this issue in point (D) of
> our response letter. In the manuscript, we have moved the figure to
> a new Appendix C, and we now emphasize more clearly that it is meant
> only as a brief illustration of a possible future study. A thorough
> assessment of its ideas would require much more testing and analysis
> in a separate paper. The focus of our present manuscript is our mathematical
> analysis. If the figure is distracting in that context, we are happy
> to remove it.

---

> ### Author Response · Authors · 2023-10-04
> **Response to Less Important Changes (Reviewer KAiw)**
>
> Section 2.1:
>
> 1) The reward and state transition functions are
> not usually defined as deterministic. This would be a big change from
> the usual setting, and if the analysis relies on this fact it could
> limit the usefulness of the paper's findings (a lot of issues in RL
> go away with deterministic rewards and transitions).
>
> 2) Similarly, the agent's policy is not necessarily
> (or even usually) deterministic. Do the authors mean that the policy
> is deterministic in the algorithms considered in the paper?
>
> 4) Much of the presentation in this section is specific
> to a subset of RL, and it would be better to be clear about this.
> E.g., $Q^{\mu}\left(s,\mu(s)\right)=V^{\mu}(s)$ is only true for
> deterministic policies.
>
> Ans: Indeed, we are concerned here with deterministic policies, rewards,
> and dynamics. We establish that setting in the first paragraph of
> Section 2.1, and we now motivate that choice in the final paragraph
> of the same section (basically, our own work deals with control issues
> surrounding these kinds of tasks in OpenAI Gym and the Deep Mind Control
> Suite, and also, this is the simplest setting where the bootstrapping
> puzzles we are addressing arise). In that same paragraph and in Section
> 4, we acknowledge that an extension of our results to a stochastic
> setting is an important goal for future work.
>
> We now mention, just after equation (2.2), that $Q^{\mu}\left(s,\mu(s)\right)=V^{\mu}(s)$ holds in a deterministic setting.
>
> 3) The discount factor can be 0 (then value becomes
> immediate reward), but not 1.
>
> Ans: We excluded the case where $\gamma=0$ because then, as you say, the
> action-value is just the immediate reward, and we don't need bootstrapping
> (because we can train the critic with supervised learning). The results
> of section 3 are essentially moot when $\gamma=0$. We did not include
> this remark in the text because this case rarely comes up in our experience.
>
> 5) It would be good to more clearly state that the
> Off-Policy Deterministic Policy Gradient theorem these algorithms
> rely on is based on an approximation (dropping the gradient of the
> action value function from the derivation entirely) that is only valid
> in the tabular setting (see Errata B in Degris et al. (2012)), and
> can be a problem when using the resulting algorithms with function
> approximation (Imani, 2018).
>
> Ans: We now acknowledge, just below equation (2.7) that the DPG theorem
> of Silver et al. (2014) is an approximation, and we cite Degris et
> al. (2012) and Imani et al. (2018). The latter paper gives a continuous-control
> example involving aliasing where this approximation leads to a suboptimal
> policy. We can mention that example if you like, but so far we have
> not, because it is not central to our aims: we take the DPG theorem
> as it stands, just as the DDPG, TD3, and SAC algorithms do, and we
> explore mathematically why Bellman-based learning leads to critics
> that produce usable approximations of $\partial Q^{\mu}/\partial a$
> despite the concerns raised by Fujimoto et al. (2022).
>
> Sections 3.1 and 3.2:
>
> 1) "This bound helps explain why DPG-based
> actor-critic learning works well in practice." It might
> be better to explicitly say why, as this was not clear to me at first.
> Is it because DPG-based AC algorithms use the BE-minimizing critic
> to estimate partial derivatives (that are shift-invariant), which
> is harmed less by the BE being a bad proxy for value error than algorithms
> that use the critic in ways that depend on low VE?
>
> 2) More context/explicit explanation would be helpful
> for the last paragraph of 3.2. Why would we want to be small? I can
> infer it's because a smaller difference error means better estimates
> of partial derivatives which means better actor updates?
>
> Ans: To answer part (1), we have modified the beginning of Section 3.1
> to better motivate the use of the differential error and clarify what
> it measures, and at the end of the same section, we now provide more
> details to answer question (2). We have also rewritten the end of
> Section 3.2 to clarify what we mean by our partial trade-off and the
> value of lowering temporal frequency.

---

### Author Response · Authors · 2023-10-04
**Revision Cover Letter**

Dear Editor and Reviewers,

Thank you for the prompt processing of our submission. Your feedback has helped us clarify several points in our work. Before answering the reviewers' individual comments, we briefly summarize the main modifications we have made.

A) Introduction and scope of the paper: We have expanded the introduction to clarify the problem we address in the paper, the purpose of the article (which is to provide theoretical results that explain certain empirical facts reported in the literature), and the setting in which we develop our results. We have also modified the last section of the paper to provide a clearer outlook on our results, as discussed in part (D) below.

B) Motivation of the difference error: Two reviewers asked for more motivation for our use of the difference error. Accordingly, we have rewritten parts of Section 3.1 and provided a new Appendix A with more details.

C) Figure 3.1, spatial factors affecting temporal frequency: As requested, we have shortened the figure caption and added several paragraphs about the figure to the main text. This revised text provides more details about the objects depicted in this figure, including the precise functions used for the reward and the policy, as well as the type of function approximator used as a critic.

D) Figure 4.1, variants of TD3: This example seems to have caused some confusion. Its purpose was to illustrate
briefly one idea of how our mathematical results might be applied in future work, and was by no means an experimental evaluation of
our results or a thorough test of a new algorithm. We have now moved it to a new Appendix C, where we more clearly articulate its purpose. If the editor and the reviewers still consider it a source of confusion, we are happy to remove the figure and accompanying text from the paper.

E) We have also rewritten section 4 and parts of other sections to clarify that this work is a theoretical paper, whose purpose is to explain why the current techniques and practices lead to the surprising performance of DPG-based actor-critic algorithms. We are well aware that there is a gap to bridge between our theoretical results and algorithm design, and we prefer to address these topics properly in future work.

Looking forward to your feedback,

Best regards,
The authors

---

### Decision · Action_Editor_JRPP · 2023-11-13

**Recommendation:** Accept as is

**Comment:**

The authors improved their submission a fair bit based on the reviewer's feedback, which helped with clarity and motivation. Most of the reviewers agree that that the theoretical results are well-motivated and of interest to researchers working with actor-critic methods. I tend to agree with them, and am recommending acceptance based on the TMLR guideline to welcome "theoretical studies yielding new insight into the design and behavior of learning in intelligent systems".

Some final comments from some of the reviewers:
*  It would have been preferrable to run the experiment and compare the magnitudes of the DFT terms of the observed Bellman Errors for each of the algorithms
*  $e_t^{\Delta Q}$ contains two parts: $\frac{\delta Q}{\delta a}$ and $\frac{\delta Q}{\delta s}$, instead of $\frac{\delta Q}{\delta a}$ alone. So it is still not quite satisfying to use $e_t^{\Delta Q}$ to justify the behavior of deterministic policy gradient.
*  Equation A.1 states that the Q-Error and the gradient-error are linearly related when ignoring non-linear effects between the inputs and outputs of the Q function. The latter assumption seems problematic, as the nonlinear effects can be large in general.
*  The claim that the Bellman error is a good proxy for minimizing the difference error might need more justification.

**Audience:**

Yes.

**Claims And Evidence:**

This paper provides a (mostly) theoretical analysis of the impact of using Bellman error for training critics in actor-critic methods, and argue that performance depends mostly on _difference error_ (a term they introduce).
The theoretical claims are well-supported, and the authors also provide some empirical results to complement them.